# A general assay platform to study protein pharmacology using ligand-dependent structural dynamics

Daniel A. Ciulla [1,3], Patricia K. Dranchak [1,3], Mahesh Aitha[1], Renier H. P. van Neer [1], Divia Shah[1], Ravi Tharakan[1], Kelli M. Wilson [1], Yuhong Wang[1], John C. Braisted[1] & James Inglese [1,2] ✉

Drug design strategies represent a fundamental challenge in chemical biology that could benefit from the development of next-generation high-throughput assays. Here we demonstrate that structural dynamic changes induced by ligand binding can be transmitted to a sensor protein fused to a target protein terminus. Here, NanoLuc luciferase, used as the intact protein or its α-complementation peptide, was fused to seven proteins from distinct enzyme superfamilies resulting in sensitive ligand-dependent bioluminescent outputs. This finding allows a general non-competitive, function-independent, quantitative, isothermal gain-of-signal ligand binding readout. As applied to chemical library high throughput screening, we can observe multivariate pharmacologic outputs including cofactor-induced synergy in ligand binding, as well as an example of allosteric site binding. The structural dynamics response assay format described here can enable the investigation of proteins precluded from study due to cost-prohibitive, insensitive, or technically challenging assays, including from cell lysates containing endogenously expressed gene edited proteins.

From seminal research surrounding the lac operon came a finding demonstrating β-galactosidase activity could be reconstituted from two fragments of the protein, a smaller alpha (α) peptide and larger omega (ω)-fragment, a phenomenon known as α-complementation[1]. In the years since, the concept of α-complementation has been applied to several proteins[2–4], including a light-generating enzyme, NanoLuc luciferase (NLuc)[5]. Such 'split proteins' have been used to create two-hybrid-like strategies for the interrogation of cellular signaling pathways with inherent drug development applications. For example, these cell-based assays have revealed how drug action may underly a cellular phenotype[6,7], whereas ligand-binding biochemical assays directly delineate the pharmacological relationship between a therapeutic target protein and its corresponding ligand chemistry[8,9].

High-throughput screening (HTS) assays that enable the efficient testing of large compound libraries and analog series are central to the medicinal chemistry that supports drug development. Depending on the nature and knowledge surrounding the macromolecular protein target, whether a receptor[10], enzyme[11], or non-enzyme, assay strategies can vary widely[12–14]. Specialized techniques or materials, such as antibodies, required to measure the interaction between a protein target and its ligand can often be incompatible with HTS and medicinal chemistry optimization. The design and implementation of an HTS platform can be complicated by target biochemistry-dependent factors, such as complex coupled-enzyme strategies[15,16], hard to obtain or unstable substrates[17–19], or HTS-incompatible post-assay derivatization steps[20]. While versatile procedures have been developed for well-

[1]Division of Preclinical Innovation, National Center for Advancing Translational Sciences, NIH, Rockville, MD, USA. [2]Metabolic Medicine Branch, National Human Genome Research Institute, NIH, Bethesda, MD, USA. [3]These authors contributed equally: Daniel A. Ciulla, Patricia K. Dranchak. ✉e-mail: jinglese@mail.nih.gov

studied protein superfamilies, such as G protein-coupled receptors[13], the diverse nature of enzymes and non-enzyme proteins appears to preclude similar generalized methodologies.

Approaches to broadly investigate target-ligand binding often carry a caveat lowering their appeal for HTS applications. For example, spectroscopically sensitive fluorophores have enabled the study of competitive ligand binding, as in fluorescence polarization (FP) assays[21]. However, FP assays require a high-affinity probe conjugated to a fluorescent dye often unavailable for novel target proteins. Moreover, the requirement of the FP probe to be nearly stoichiometrically target-bound tends to lower detection sensitivity for competing ligands. Conversely, 'label-free' detection systems, as in surface plasmon resonance (SPR) spectroscopy, make possible direct measurement of ligand interactions without a functional output or labeled ligand, although the protein target itself requires tethering to a microfluidic chip or other biosensor surface reducing throughput[22]. Today, numerous assay configurations utilizing a range of detection technologies exist. However, none are widely applicable to a broad range of biological targets in low volume HTS format.

Here, we report the discovery that ligand binding to its target protein can be detected through accompanying perturbations in the luminescence output of N- or C-terminal NLuc fusions[23]. We hypothesize that upon ligand binding, structural changes, vibrational motion arrest, or a combination of both are transmitted to the NLuc sensor protein to affect luminescence yield in an as yet undetermined manner. Further, this phenomenon can be likewise observed using self-assembling split NLuc α-complementation. Here, the high-affinity 11-mer α-peptide, HiBiT fused to either terminus of the target protein can be combined with its ω-fragment, LgBiT to report on ligand binding[24]. We have developed this finding into the platform we call the structural dynamics response (SDR) assay (Fig. 1a, b). We explore the generality of the SDR assay by measuring ligand-dependent modulation of NLuc bioluminescence in seven target proteins representing structurally and functionally diverse enzyme classes as purified enzymes or in lysates from gene edited cells.

## Results

### Ligand binding couples to sensor protein output

As a proof-of-concept target, we appended the sensor protein or corresponding α-peptide to either the N- or C-terminus of the target protein firefly luciferase (FLuc), an ATP cofactor-dependent monooxygenase (Supplementary Fig. 1a) having a broad, well-characterized and accessible ligand pharmacology[25]. For initial characterization of the SDR concept, either NLuc or the 11 amino acid HiBiT tag (α-peptide) and complementary 18 kDa LgBiT subunit (ω-fragment), which self-assemble to reconstitute functional NLuc enzyme[24] were tested. In contrast to the ATP-dependent bioluminescent FLuc enzyme, NLuc generates a bright bioluminescence in an ATP-*independent* manner from the direct oxidation of furimazine (Supplementary Fig. 1b). Therefore, these SDR constructs allow us to measure ligand binding to FLuc without a reliance on FLuc enzyme activity. Using bacterially expressed and purified FLuc-C-NLuc, FLuc-N-HiBiT or FLuc-C-HiBiT we first demonstrated the well-established potent inhibition of enzymatic activity, that is FLuc bioluminescence (Fig. 1c, d, solid circles), using PTC124, an aryl carboxylate that undergoes tight-binding to FLuc upon ATP-adenylation by the enzyme (Supplementary Fig. 1c)[26]. Next, FLuc-C-NLuc, FLuc-N-HiBiT or FLuc-C-HiBiT were assayed with the same inhibitor in an SDR assay format by monitoring bioluminescence from NLuc or its reconstitution by α-complementation with LgBiT. Measurements of NLuc activity were made either in the absence or presence of the FLuc ATP cofactor, as NLuc bioluminescence is ATP-*independent*. These circumstances permitted both an ATP-independent low affinity (Fig. 1c, open blue symbols) and ATP-dependent high affinity (Fig. 1c, solid blue symbols) binding of PTC124 by FLuc to be directly detected. For example, using FLuc-C-

HiBiT (Table 1) we observe a single pIC$_{50}$ of 7.95 ±0.02 for PTC124 in the conventional FLuc enzyme inhibition assay, whereas the corresponding pSDR$_{50}$ values are 8.30 ±0.07 and 5.73 ±0.13, with and without ATP, respectively. An additional unique feature of the SDR assay is a gain-of-signal output accompanying ligand binding, opposite to the loss-of-signal typical for inhibitors of an enzymatic assay.

**Mechanistic SAR from the SDR assay using FLuc-C-HiBiT.** From the PubChem data repository (AID 2309) we assembled a library of 1343 FLuc inhibitory compounds showcasing a diverse range of chemotypes, potencies, and mechanisms (Supplementary Data 1 and 2). The comparative 11-point quantitative HTS (qHTS)[27] outputs from both the FLuc enzyme assay and the SDR assay (+ ATP condition) are shown in Fig. 1e highlighting the mirrored and varied efficacies displayed in the SDR assay compared to inhibition of a functional enzyme activity output. Chemotype hierarchal clustering based on Tanimoto (TT) similarity score above 0.8 defined 29 clades (A-C′, Supplementary Fig. 2a–c and Supplementary Data 2) with corresponding heatmaps for the assay IC$_{50}$ or SDR$_{50}$ values as shown in Supplementary Fig. 2d.

**ATP cofactor-dependent ligand binding.** Salient features of this comparative assay screening data set include the chemotype dependence on ATP (*e.g.*, Fig. 1f, g), clades F, S, T and A′), and well-defined ATP-independent SAR of clades O and P. Compounds in clade P likely occupy the FLuc ATP binding site, supported in part, by the close overlap of the ±ATP SDR assay output-driven concentration response curves (Fig. 1f, clades O and P), and particularly in the structural resemblance of the pyrazolo pyrimidine core (clade P) to the adenine heterocycle of ATP. This pyrazolo pyrimidine contains three points of variation delineating 40 compounds, where the SDR response tracks closely with the functional enzyme assay (Supplementary Fig. 3). The binding site for the ATP-independent SDR assay response associated with the clade O chemotype, however, is less clear as this structure bears some similarity to the luciferin substrate of FLuc (Supplementary Fig. 4).

Among the ATP-dependent SDR assay responses are the aryl carboxylate-containing oxadiazoles (27 aryl carboxylates of the 35 clade T members), (Supplementary Data 1) reminiscent of adenylate-forming PTC124[26]. Additional chemotypes demonstrating an ATP-dependence on binding are shown in clades F, T and A′ members of which can be found to contain aryl (*e.g.*, clade T) or alkyl carboxylates (*e.g.*, clade F).

An interesting category of SDR concentration response curve (CRC) profiles is observed in several clades, including in clade S, where SDR assay output is strictly ATP-dependent rather than having an ATP affinity shifting effect (*e.g.*, NCGC00449795, Fig. 1f). We examined this clade of 48 compounds further, first expanding the series by 88 available analogs, and testing in the FLuc-C-HiBiT and FLuc-C-NLuc SDR assays (Supplementary Data 3). The ATP-dependence strictly held up, and within the expanded SAR we also identified inactive analogs (Supplementary Fig. 5). Aided by this SAR we conducted a Thermofluor analysis on representative active and inactive compounds using firefly luciferase (FLuc) and Nano luciferase (NLuc) enzymes. In agreement with the SDR results, thermal denaturing experiments demonstrated the compounds bound only to FLuc and in an ATP-dependent manner (Supplementary Fig. 6). Thus, the SDR assay can reveal mechanistic subtleties such as multivalent or synergistic binding, that would otherwise be difficult to discriminate in assays based on substrate turnover.

**Generality of the SDR assay**
Expanding beyond the FLuc proof of concept, we investigated several protein families, including kinases, isomerases, reductases, and ligases. We demonstrate that the SDR assay concept is general and can be applied to a target in crude eukaryotic cellular lysate allowing the

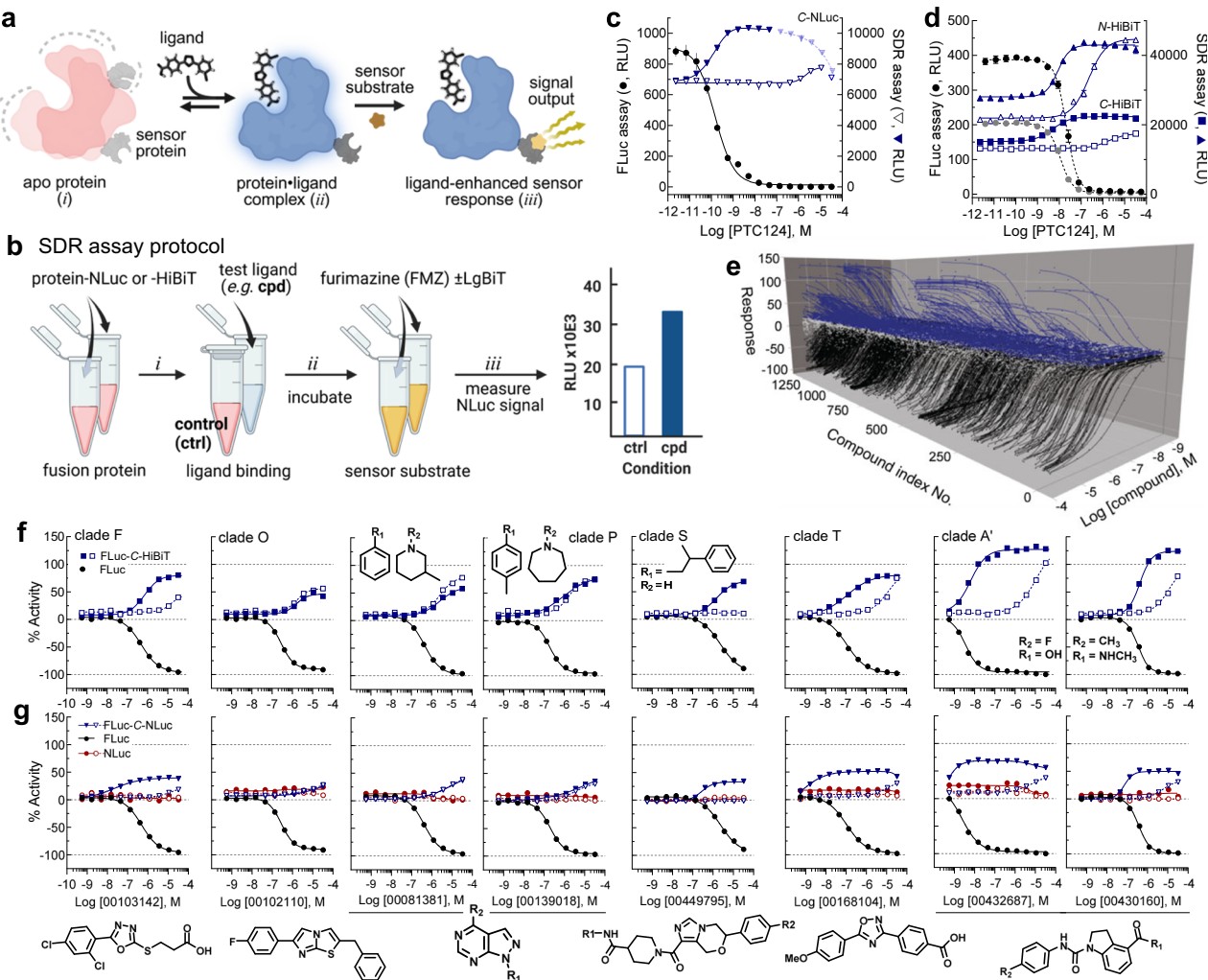

**Fig. 1 | SDR assay concept and implementation. a** Conceptual framework for an SDR assay using terminal fusion protein of interest. The equilibrium between apo (*i*) and ligand-bound protein (*ii*) can be distinguished by the sensor (*iii*) activity, where a reddish color represents a dynamic apo protein relative to a more static blue ligand-bound protein. Created in BioRender. Inglese, J. (2025) https://BioRender.com/26yvpbw. **b** Key steps in an SDR assay. In this study the assay was performed in a 7 μL volume (4 μL target protein/ligand plus 3 μL detection reagent) permitting 1536-well plate quantitative high-throughput screening, qHTS (see Methods for full protocol). **c** SDR assay proof-of-concept using a *C*-terminal NLuc fusion to firefly luciferase (FLuc-*C*-NLuc, 1 nM), or **d** an *N*- or *C*-terminal fusion of HiBiT to FLuc (FLuc-*N*- or -*C*-HiBiT, 10 nM). The left axes indicate RLU generated by the enzyme activity of FLuc with increasing concentrations of FLuc inhibitor PTC124, for a *C*-terminal NLuc fusion (**c**, black circles), or an *N*- (black circle) or *C*- (gray circle) HiBiT fusion (**d**) protein. The right axes indicate ligand-dependent SDR RLU, measured from NLuc (**c**) FLuc-*C*-NLuc (inverted blue triangle) or

reconstituted NLuc (**d**) FLuc *N*- (blue triangle) or *C*- (blue square) HiBiT, with increasing PTC124 concentration in the presence (solid symbols) or absence (open symbols) of 10 μM ATP. Error bars are SD, *n* = 6 experiments. **e** qHTS waterfall plot illustrating concentration response curves (CRCs) obtained for a library of 1,343 compounds enriched for FLuc inhibitor chemotypes. Black CRCs from FLuc enzyme assay and blue CRCs from the 10 nM FLuc-*C*-HiBiT SDR assay (+ ATP condition). **f** Representative CRC from (**e**) for indicated clades, where symbols are activity from: SDR, plus ATP (blue solid square), SDR, no ATP (blue open square), and FLuc enzyme (black circle) assays. For full SAR around clade P see Supplementary Fig. 3. **g** Comparative SDR activity of clade representative compounds using the 1 nM FLuc-*C*-NLuc protein with ATP (blue solid triangle), no ATP (blue open triangle), or 1 nM NLuc without ATP (open red circle), or with ATP (solid red circle). Supplementary Data 1, 2 support (**e**–**g**). RLU, relative light units; No., number; Cpd., compound. Source data are provided in a Source Data file.

**Table 1 | Binding of PTC124 ± ATP from functional vs SDR assays**

| ±ATP (μM) | FLuc-*C*-NLuc (1 nM) | | FLuc-*N*-HiBiT (10 nM) | | FLuc-C-HiBiT (10 nM) | |
|---|---|---|---|---|---|---|
| | pIC$_{50}$ | pSDR$_{50}$ | pIC$_{50}$ | pSDR$_{50}$ | pIC$_{50}$ | pSDR$_{50}$ |
| 10 | 9.85 ± 0.03 | 9.95 ± 0.05 | 7.68 ± 0.02 | 8.07 ± 0.05 | 7.95 ± 0.02 | 8.30 ± 0.07 |
| 0 | NA | 5.88 ± 0.04 | NA | 6.64 ± 0.05 | NA | 5.73 ± 0.13 |

Error represents SD, *n* = 6; NA, no activity, FLuc enzyme activity is ATP-dependent. SDR output based on NLuc is ATP-independent. Source data are provided as a Source Data file.

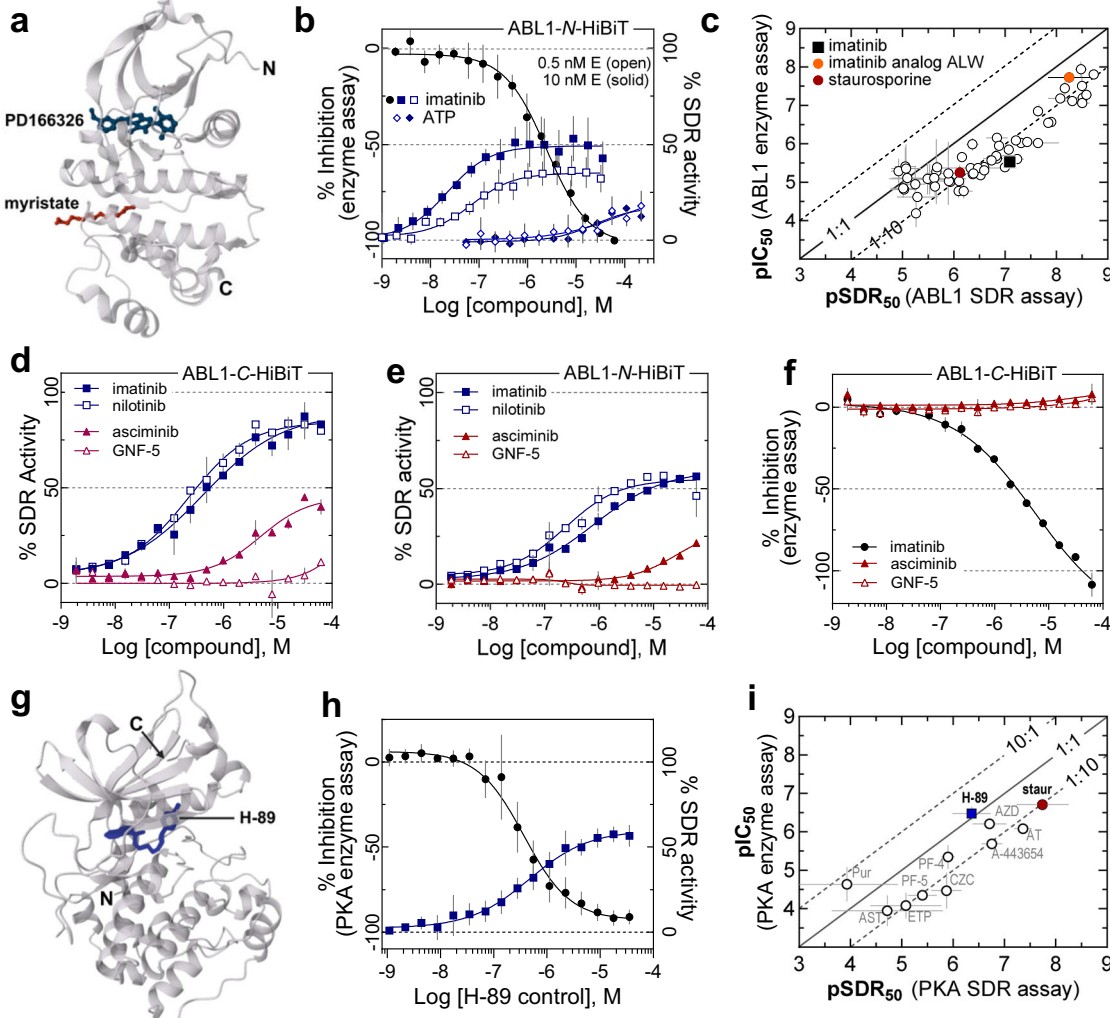

**Fig. 2 | SDR protein kinase assay. a** ABL1 kinase domain bound to PD166326 inhibitor (dark blue) and myristic acid (red) (PDB: 1OPK), Nagar B et al. **b** SDR output of imatinib (blue square) or ATP (blue diamond) binding to 0.5 nM (open symbol) or 10 nM (solid symbol) ABL1-*N*-HiBiT kinase domain or imatinib inhibition of kinase activity (black circle) using Kinase-Glo Plus reagent (KGP). Error bars represent the SD, n = 3 experiments (Imatinib SDR) or $n = 4$ experiments (ATP SDR and imatinib kinase activity). **c** Correlation plot comparing the ABL1 KGP assay pIC$_{50}$ vs pSDR$_{50}$ (10 nM enzyme without ATP) for all inhibitors identified from the 128-member kinase inhibitor library. Colored symbols reference compounds discussed in Results section. For underlying data see Supplementary Data 4, 5, and Supplementary Fig. 9. **d, e** SDR output for myristate site ligands asciminib (red solid triangle) and GNF-5 (red open triangle) compared to imatinib (blue square) and

nilotinib (open square) for ABL1-*C*- or -*N*-HiBiT, respectively. **f** Effect of allosteric ligands on enzyme catalysis (imatinib, black circle, asciminib, solid red triangle, and GNF-5, open red triangle). Error bars for **d**, **e**, and **f** represent the SEM, $n = 2$ technical replicates. **g** Protein kinase A (PKA) bound to H-89 inhibitor (dark blue) (PDB: 1YDT), Engh RA et al. **h** PKA enzyme activity (black circle) and SDR assay (blue square) CRC for H-89. Error bars represent the SD, $n = 3$ experiments **i** Correlation plot comparing the PKA KGP assay pIC$_{50}$ to PKA SDR assay pSDR$_{50}$ (without ATP). H-89 (blue square) and staurosporine (staur., red circle) indicated. See Supplementary Data 6 and Supplementary Fig. 11 for underlying data and compound information. Supplementary Data 4–6 supports panels (**c**, **i**). Source data are provided in a Source Data file.

study of proteins not easily accessible from bacterial overexpression systems. The first examples illustrate the SDR assay applied to protein kinases in an ATP- and competitive ligand probe-independent format.

**Abelson tyrosine kinase.** (ABL1), first identified as the BCR–ABL1 fusion protein in patients with Philadelphia chromosome-positive human leukemia, has been the target of successful chronic myeloid leukemia therapeutics[28,29]. An N-terminal HiBiT ABL1 kinase domain (Fig. 2a, D233-S500[30]) was expressed and purified from *E. coli*[31]. A limit-of-detection of basal RLU demonstrated sub-nanomolar sensitivity (Supplementary Fig. 7, *black bars*). The functional kinase activity was confirmed using a Kinase-Glo Plus (KGP) luminescence assay to measure tyrosine phosphorylation of the abltide peptide (KKGEAIYAAPFA-NH$_2$) substrate through ATP depletion and subsequent kinase inhibition by imatinib[32] (Fig. 2b, *solid circles*). An

imatinib-binding gain-of-signal SDR assay response was demonstrated with concentrations of ABL1-*N*-HiBiT between 0.5 and 10 nM (Supplementary Fig. 7, *blue bars*). From an imatinib titration, the SDR assay was directly compared to the functional KGP assay (Fig. 2b). Here, we observe an ATP-dependent 27-fold rightward imatinib potency shift (pIC$_{50}$ = 5.60 ± 0.15 versus pSDR$_{50}$ = 7.03 ± 0.17) in the enzyme assay versus the SDR assay format, respectively, illustrating the inherently greater sensitivity in the absence of a competing nucleotide cofactor. Moreover, we can directly measure the ATP affinity by the SDR assay (pSDR$_{50}$ = 4.61 ± 0.27), which agrees with reported $K_m$ values[33,34] (Fig. 2b, *blue diamonds*).

pIC$_{50}$ values obtained with the functional enzyme assay were next compared to the pSDR$_{50}$ across a 128-compound library of kinase inhibitors enriched for those annotated as having ABL1 as a target kinase (Supplementary Fig. 8 and Supplementary Data 4, 5). CRCs for

each condition primarily gave gain-of-signal SDR assay responses upon ligand binding, as with imatinib and with the pharmacology observed in the FLuc case (Supplementary Data 5 and Supplementary Fig. 9). For ABL1-*N*-HiBiT the SDR assay maximum response, ~50% activity relative to vehicle only control, is not as pronounced as that observed for FLuc-*C*-HiBiT. This may reflect the degree to which the structural dynamics response can transfer from target protein to sensor.

Correlated inhibitor potencies obtained from the functional enzyme versus SDR assays illustrate, aside from a few exceptions, an equivalence or apparent higher sensitivity of the gain-of-signal SDR assay format (Fig. 2c). Assay sensitivity or robustness will be, in part, based on factors such as the quantifiable response for a specific target protein concentration and the ligand binding mechanism, *e.g.*, ortho- vs allosteric site binding. Also, target protein concentrations may exceed the $K_D$ of potent ligands limiting the observable affinity (*e.g.*, see Table 1). This could, for example, compress the potency ranking in a medical chemistry optimization series, but is unlikely to impact discovery of nascent unoptimized compounds typically having $K_D$ values well above the low nM target protein assay concentration[12,35].

The predictive power of the SDR assay to identify kinase inhibitors is shown, for example, by the 4-methyl-N-phenylbenzamide, ALW-II-41-27 (ALW) (Supplementary Fig. 8). While not annotated in the Pharos database as an ABL kinase inhibitor, ALW was identified as a potent inhibitor, a result supported by a structure containing elements common to the ABL1 kinase inhibitor imatinib.

Within the library were two micromolar affinity myristate-site binders, GNF-2 and GNF-5[36]. While neither was identified by either functional enzyme or SDR assays (Supplementary Fig. 8, 9), asciminib, an optimized and FDA-approved myristate pocket ligand, was detected by both the ABL1-*N*-HiBiT and ABL1-*C*-HiBiT (Fig. 2d, e), with the latter SDR construct beginning to detect GNF-5 (Fig. 2e). Our functional enzymatic ABL1 kinase assay was unable to detect asciminib (Fig. 2f and Supplementary Fig. 10) as previously reported[36]. Detection of GNF-5 may reflect the location of the *C*-terminal sensor in the vicinity of the myristate-binding pocket (Fig. 2a)[37]. These data illustrate that the SDR assay has the capability to detect ligands in auxiliary binding sites outside of the active site, suggesting a means to measure allosteric binding in a straightforward high throughput manner.

**Protein Kinase A.** (PKA), the first protein kinase structurally elucidated[38], is a cAMP-regulated molecular switch responsible for mediating numerous cellular functions and signal transduction pathways through serine/threonine phosphorylation. Similar to the ABL1 *N*-terminal α-peptide fusion protein, PKA-*N*-HiBiT displayed a gain-of-signal SDR assay response upon binding the isoquinoline sulfonamide H-89[39], an ATP site binding PKA inhibitor (Fig. 2g, h)[40]. The sensitivity of the functional assay measured using the kemptide peptide (LRRASLG-NH₂), versus the SDR assay can vary depending on the ligand, as observed for H-89 ($pIC_{50} = 6.45$; $pSDR_{50} = 6.67$) compared to staurosporine ($pIC_{50} = 6.67$; $pSDR_{50} = 7.71$, Fig. 2i). However, from an interrogation of the 128-member ABL1 kinase-enriched inhibitor library with PKA-*N*-HiBiT protein we observed that the SDR assay format displays a higher general sensitivity compared to the functional enzymatic assay (Fig. 2i) as was observed for ABL1 kinase.

As the library was enriched for ABL1 kinase inhibitors and therefore biased toward tyrosine kinases, as expected, there was a lower overall activity, 23% versus 56% for inhibitors of PKA versus ABL1, respectively (Supplementary Data 5, 6 and Supplementary Figs. 9 and 11). While low, the inhibitory activity toward PKA is higher than the annotated frequency of PKA inhibitors of 7% in the library. Of the nine compounds annotated as PKA inhibitors, five were identified in both the functional kinase and SDR PKA assays, while 22 additional compounds were detected by the SDR assay, five of which were also inhibitors in the KGP assay format (Supplementary Fig. 11).

**Co-factor independent phosphoglycerate mutase.** (iPGM), an isozyme of the human glycolytic enzyme, catalyzes the interconversion of 2- and 3-phosphoglycerate. Because iPGM bears no sequence identity to the mammalian cofactor-dependent form, dPGM, it has been considered a drug target for diseases caused by infectious organisms, such as parasitic nematodes[16]. The conventional approach to measuring iPGM activity requires a coupled assay involving two additional enzymes to convert the 2-phosphoglycerate product to ADP and phosphoenolpyruvic acid (PEP), then to ATP and pyruvic acid. A third enzyme, firefly luciferase can use ATP to generate a bioluminescence output. Previously, we developed ipglycermides, a class of high affinity cyclic peptide ligands that bind between the phosphotransferase and phosphatase domains of nematode iPGMs (Fig. 3a)[41]. Using the *C. elegans* iPGM we created a *C*-terminal HiBiT iPGM fusion protein that maintains a 50-fold signal-to-background down to 50 pM enzyme concentration when complemented by the LgBiT ω-fragment (Supplementary Fig. 12) and demonstrates saturable SDR assay response when bound to the cyclic peptide inhibitor, ipglycermide Ce-2 (Fig. 3b). We examined the limit of detection obtainable for the high affinity ipglycermide Ce-2, determined by surface plasmon resonance kinetics to have a 38 pM $K_D$[16], using either the enzyme-coupled assay or the SDR assay. Here, we can observe a CRC in the SDR assay format for Ce-2 using as little as 20 pM iPGM-*C*-HiBiT, 16-fold below that needed in the iPGM enzymatic assay (Fig. 3c).

The iPGM-HiBiT SDR assay was used to compare $pSDR_{50}$ for a set of ipglycermide analogs against the $pIC_{50}$ obtained from an FP assay based on a fluorescein-conjugated ipglycermide probe, Ce-2d-FL. The resulting correlation plot similarly ranks the analog potencies between both assays. However, the higher affinity ipglycermides, Ce-2, Ce-2 P4V and Ce-2 Y7F (Fig. 3d, red symbols) deviate from the linear relationship. This latter observation reflects the higher sensitivity of the SDR assay that can be conducted at lower enzyme concentrations for ipglycermides having $K_D$ below the iPGM concentration needed in the fluorescent polarization assay. Additionally, the ability of the SDR assay to determine ortholog-selective affinity of these ipglycermide analogs was tested. The higher potency of the Ce-2 ipglycermide series for *C. elegans* versus *B. malayi* iPGM is faithfully recapitulated using a comparative SDR assay format for both orthologs (Fig. 3e).

**DNA ligases.** (Lig) catalyze the repair of DNA breaks in either an NAD⁺ or ATP-dependent reaction[42]. Using DNA ligases from *E. coli* and bacteriophage T7 (Fig. 4a), *N*- and *C*-terminal HiBiT-containing enzymes were recombinantly expressed and purified from *E. coli*[43,44]. The enzymatic function of each ligase was ascertained using a standard gel electrophoresis-based break repair procedure where *Hind*III-digested λDNA was treated with the respective ligase and appropriate cofactor[45]. All ligases demonstrated DNA repair in their optimized buffer conditions (Fig. 4b). The *N*-terminal HiBiT *E. coli* and *C*-terminal HiBiT T7 ligases showed a robust gain-of-signal SDR assay response using a synthetic 22-mer double stranded DNA oligonucleotide, the corresponding *C*- and *N*-HiBiT proteins resulted in an opposite, though less pronounced response (Fig. 4c, d). This observed assay response variation, in both magnitude and direction, may be a consequence of DNA-mediated domain motions to either expose or obscure the terminal α-peptide sequence, or affect the split NLuc active conformation.

The DNA ligase from *E. coli* uses β-nicotinamide adenine dinucleotide (NAD⁺) as a cofactor[42], while the T7 DNA ligase is ATP-dependent. With the appropriate adenyl cofactors we were able to monitor significant and specific SDR responses to NAD⁺ and ATP, for *E. coli* and T7 DNA ligase, respectively (Fig. 4e–h). The nucleotide $SDR_{50}$ values measured in these experiments fall between ~1–100 μM (Supplementary Data 1), in line with published values[46,47]. The *E. coli* *N*-HiBiT ligase robustly responded to its NAD⁺ cofactor, however the *C*-HiBiT construct did not respond to cofactor binding (Fig. 4e).

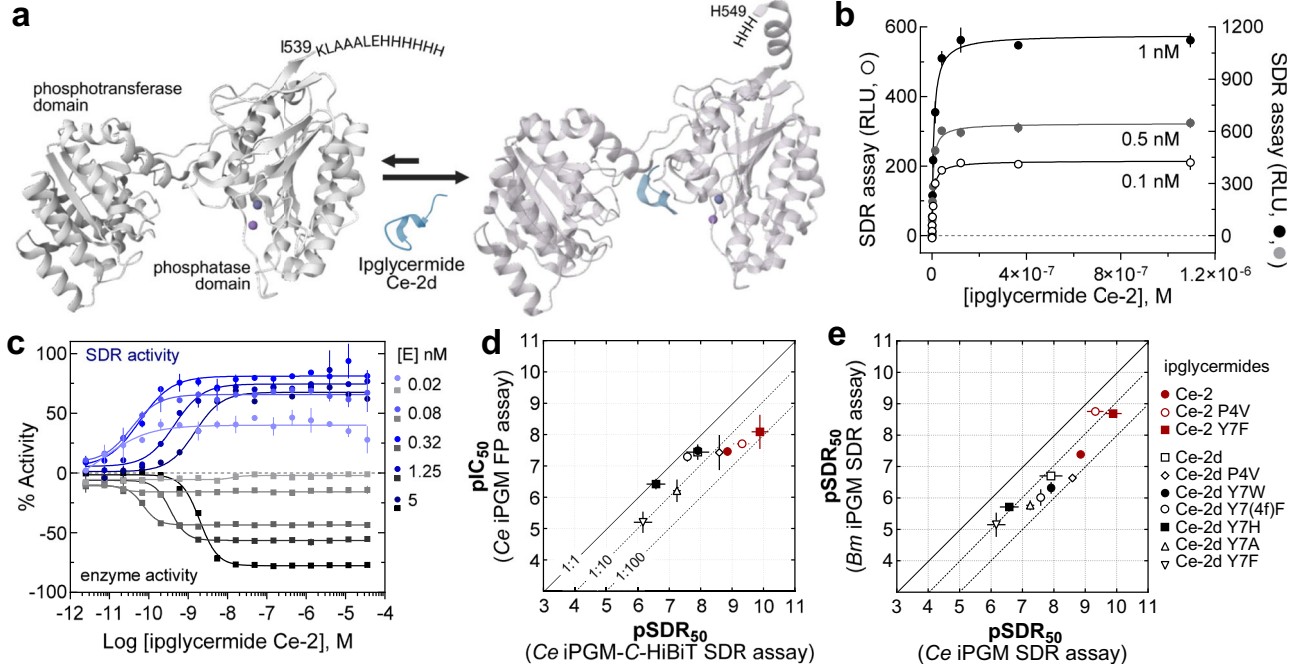

**Fig. 3 | Ligand selectivity and affinity for cofactor-independent phosphoglycerate mutase (iPGM) using SDR. a** Crystal structures of the apo (PDB 5KGL) and ipglycermide Ce-2d bound (PDB 5KGN) iPGM from Yu H et al. Metal ions are identified as purple spheres. **b** Saturation binding of Ce-2 to 1 nM (black circle) 0.5 nM (gray circle) or 0.1 nM iPGM (open black circle). **c** Relative detection sensitivity of the SDR assay (blue circle) compared to a functional couple-enzyme assay (gray/black square) for ipglycermide Ce-2 binding *C. elegans* iPGM. Error bars for (**b**, **c**) represent the SEM of 2 experiments. **d** Correlation plot comparing the binding potencies for ipglycermide analogs from either an FP-based competition binding (10 nM iPGM) or SDR assay (0.5 nM iPGM-*C*-HiBiT) format. The FP assay uses a fluorescein-labeled Ce-2d analog (Ce-2d-FL). **e** Correlation plot comparing the binding potencies for ipglycermide analogs between *B. malayi* and *C. elegans* iPGM orthologs using the SDR assay. Ipglycermides are identified by symbols to the right. Error bars for (**d**, **e**) represent the SD, $n = 3$ experiments. RLU, relative light units; FP, fluorescence polarization; Ce, *C. elegans*; Bm, *B. malayi*. Source data are provided in a Source Data file.

In contrast T7 DNA ligase was silent with $NAD^+$, as expected, whereas ATP binding resulted in a strong SDR signal for both the *N*- and *C*-HiBiT T7 ligases (Fig. 4f, h). We can attribute this termini-specific response to the larger size and more distant *C*-terminus of *E. coli* DNA ligase from its *N*-terminally located $NAD^+$ binding site (Fig. 4a, *top left*). In the smaller T7 ligase both *N*- and *C*-termini are proximal to the ATP binding site (Fig. 4a, *top right*), possibly explaining the observed SDR in both T7 ligase constructs (Fig. 4h).

Interestingly, ATP showed a weak though reproducible SDR output on *N*-terminal HiBiT *E. coli* ligase (Fig. 4g). Given that $NAD^+$ is an adenyl cofactor containing the adenosine diphosphate component of ATP (Fig. 4a), this observed SDR signal may reflect some level of ATP recognition by the DNA ligase.

**Dihydrofolate reductase.** (DHFR) catalyzes the NADPH-mediated hydride transfer to dihydrofolate (DHF) to generate tetrahydrofolate (THF), a central reaction of one-carbon metabolism and de novo synthesis of purines, pyrimidines, amino acids, and the target of the anticancer drug methotrexate, a folate analog[48]. The field has primarily been using the original DHFR assay described in 1958[49], that measures the 340 nm absorbance decrease accompanying NADPH oxidation to $NADP^+$. The assay has significant HTS and general lab use shortcomings due to the anaerobic requirements of the substrate DHF[19]. From a SDR assay using *C*-terminally HiBiT tagged human DHFR (Fig. 5a)[50] we measured the respective low- and high-affinity saturation binding of methotrexate (MTX) across DHFR concentrations at 0.5 or 5 nM in either the absence or presence of 5 μM NADPH redox cofactor that give the expected $K_D$ values (Fig. 5b). CRCs, as shown in Fig. 5c for the NADPH-dependence on MTX $SDR_{50}$, were also determined from a titration of MTX (pM–μM), which demonstrated the expected increase in potency with increasing NADPH concentration[51].

A comparative analysis between a standard DHFR enzyme assay that measures the increase in absorbance accompanying NADPH reduction of DHF and turnover to $NADP^+$ illustrates the sensitive nature of the SDR assay format. Here, six antifolates having a range of DHFR affinities were compared using the DHFR-*C*-HiBiT enzyme in the functional assay and the SDR assay. The antifolates, aminopterin (A), methotrexate (M), γ-fluoromethotrexate (F), pralatrexate (Pr), and trimetrexate (T), showed ~100-fold enhanced sensitivity in the SDR assay format while pemetrexed (Pm), showed poor DHFR affinity in both assays (Fig. 5d). The observed higher assay sensitivity in the SDR assay format is primarily due to the lower DHFR concentrations and non-competitive nature of this format, allowing the apparent affinities to more closely approximate the ligand's $K_D$.

**SDR assay using gene edited endogenous proteins from mammalian cell lysates.** The sensitivity of the SDR assay suggests that this method may be applied to endogenously expressed proteins obtained from, for example, mammalian cellular extracts. To test this, we edited the 3′-end of the endogenous *DHFR* gene in HEK293 cells to include the HiBiT sequence using CRISPR/Cas9 mediated homologous recombination[52,53]. Cell lysates were characterized for HiBiT insertion by Western and Nanoblot analysis that confirmed expression of the expected DHFR-*C*-HiBiT fusion protein (Supplementary Fig. 13). Using a 100-fold lysate dilution, which we estimate equates to a DHFR-*C*-HiBiT assay concentration of ~0.2 nM, we measured the SDR output for a MTX titration in the absence or presence of added NADPH (Fig. 5e). Here, we observed the expected leftward $SDR_{50}$ shift for MTX in the

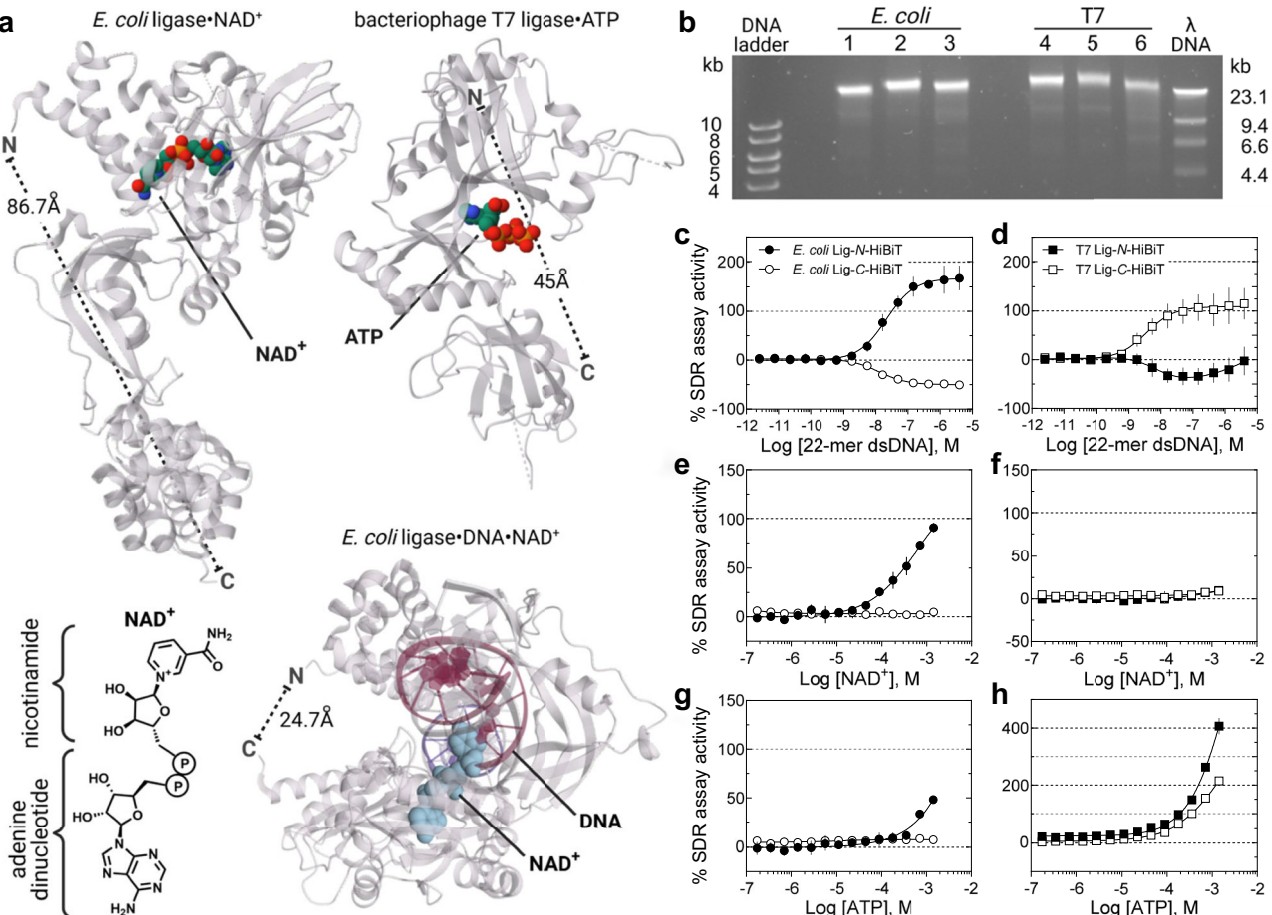

**Fig. 4 | Nucleic acid and cofactor-specific binding to DNA ligase measured using the SDR assay. a** DNA ligase crystal structures illustrating the relative positions of the *N*- and *C*-termini, DNA and nucleotide binding sites for the *E. coli* ligase (MW 75 kDa) complexed with NAD[+] (PDB 5TT5) from Unciuleac MC et al., with NAD[+] and DNA (PDB 2OWO) from Nandakumar J et al., and bacteriophage T7 ligase (MW 41.1 kDa) complexed with ATP (1A0I) from Subramanya HS et al. The structure of NAD[+] is shown. **b** Agarose gel electrophoresis of *Hind*III-digested λDNA repair by ligases from *E. coli* and bacteriophage T7 containing *N*- or *C*-terminal HiBiT α-peptide. Ligase concentrations used: 1, NEB *E. coli* ligase, 0.5 U/μL; 2, *E. coli N*-HiBiT,

1 μM; 3, *E. coli C*-HiBiT, 100 nM; 4, NEB T7 ligase, 150 U/ μL; 5, T7 *N*-HiBiT, 100 nM; 6, T7 *C*-HiBiT, 1 μM. Control ligases were from NEB and used as directed. Gel is representative of 2 replicates. Concentration-response curves obtained for the SDR assay for 0.5 nM *E. coli* Lig-*N*-HiBiT (solid circle), 1 nM *E. coli* Lig-*C*-HiBiT (open circle) and 1 nM T7 Lig-*N/C*-HiBiT (solid/open squares, respectively). **c**, **d** a 22-mer dsDNA oligo; **e**, **f** NAD[+]; **g**, **h** ATP. Error bars are the SD, *n* = 3 experiments. Supplementary Table 1 supports panels (**c**–**h**). kb, kilobase; Lig, ligase. The uncropped gel and source data are provided in a Source Data file.

presence of NADPH, and this NADPH-dependent potency extended to the antifolate panel examined earlier (Fig. 5f).

## Discussion

Using the SDR output, we measured concentration-dependent responses for a variety of enzymes and their corresponding ligands, ranging from small molecules and peptides to macromolecules such as DNA. We've demonstrated how the SDR assay can consolidate disparate functional assays into a single straightforward format using bioluminescence generated from either full length NLuc or NLuc reconstituted by α-complementation. While other split-reporter systems are available[3,54], the sensitivity, low $K_D$, and rapid onset of HiBiT/LgBiT complementation allows for the study of target proteins in the nM to sub-nM concentration range. Importantly, the $SDR_{50}$ we measure for these interactions are in general agreement with previously published work (*i.e.*, ligand potency, $IC_{50}$ or affinity, $K_D$), further validating the use of the SDR assay as a quantitative approach.

Although the underlying mechanism by which the SDR assay can reveal subtle pharmacological features of ligand binding remains undetermined, we speculate a combination of structural rearrangements and vibrational dampening of the protein contributes to the ligand-dependent signal. Ligand-induced movement of protein termini

can result in significant disorder/order transitions or spatial reorientations. Evidence supporting ligand-mediated secondary structure formation is captured in a pair of *C. elegans* iPGM crystal structures where the disordered *C*-terminus of the apo enzyme adopts an α-helix in the Ce-2d bound complex (Fig. 3a), while in the *E. coli* DNA ligase the positions of the *N*- and *C*-termini converge 62 Å upon DNA binding (Fig. 4a). More broadly, the compelling comparative functional and SDR assay-derived SAR obtained with FLuc-*C*-HiBiT or FLuc-*C*-NLuc suggest the specific ligand binding interactions correlate with split NLuc or intact NLuc luminescence, respectively (*e.g.*, Supplementary Fig. 3–5). Additionally, the SDR observed in DHFR-*C*-HiBiT for antifolates (Fig. 5d) is consistent with the coalescing of solution conformers upon MTX binding (Fig. 5g, h), which may link to the NLuc catalytic site by, according to Hammes-Schiffer and Benkovic, a network of coupled motions throughout the protein fold that facilitate the chemical reaction[55].

In contrast to cell-based reporter assays, where ligand-mediated gene expression or ligand-stabilized protein turnover[56] can modulate protein levels[52,53], it has not been appreciated to our knowledge that a protein ligand binding event could substantially and directly modulate a sensor protein fusion output. While both thermal-shift (*e.g.*, ThermoFluor) or cellular thermal shift assay (CETSA)-based approaches

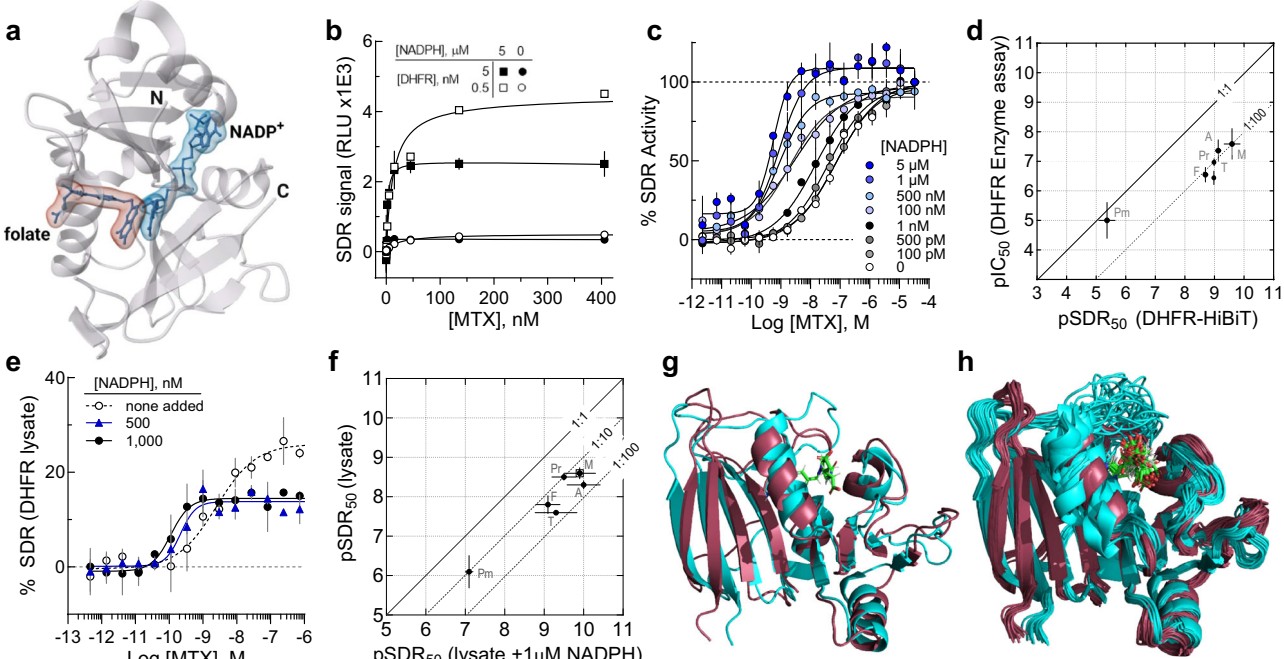

**Fig. 5 | Ligand binding observed from purified recombinant DHFR or a mammalian cell lysate containing endogenous gene edited DHFR using the SDR assay. a** DHFR bound to folate (red molecular surface) and NADP⁺ (blue molecular surface) (PDB 4M6K) from Bhabha G et al. with *N*- and *C*-termini indicated. **b** Co-factor dependent SDR saturation binding curves for methotrexate (MTX) binding to 5 nM (solid symbol) and 0.5 nM (open symbol) DHFR-*C*-HiBiT in the presence (square) or absence (circle) of saturating NADPH. **c** SDR assay concentration response curves for MTX binding to 0.5 nM DHFR-*C*-HiBiT for various [NADPH]. Data normalized to 32.5 μM MTX response. **d** Correlation analysis of antifolate chemotherapeutic affinities as determined by a functional DHFR-*C*-HiBiT assay (100 nM, 75 μM NADPH) versus SDR assay using DHFR-*C*-HiBiT (0.5 nM, 5 μM NADPH), respectively. **e** SDR assay concentration response curves for MTX binding in 1:100 cellular lysate from DHFR *C*-terminus HiBiT gene edited HEK293 cells for indicated NADPH concentrations. Data normalized to vehicle control. **f** Correlation

analysis of antifolate affinities determined by the SDR assay for a 1:100 lysate (~ 0.2 nM enzyme) without added NADPH or with 1 μM NADPH, respectively. The antifolates tested were methotrexate (M), γ-fluoromethotrexate (F), pralatrexate (Pr), aminopterin (A), trimetrexate (T), and pemetrexed (Pm). Error bars represent the SEM, $n = 2$ technical replicates, representative of 2 (**b**) or 3 (**c**) experiments, or SD, $n = 3$ experiments (**d**, **e**, and **f**). **g** Aligned minimal energy solution NMR conformer structures of apo (cyan, PDB: 2L28) and holo MTX-bound (raspberry, PDB: 1AO8) *L. casei* DHFR illustrating the conformational shift induced by the association of MTX (green). **h** Aligned solution NMR conformer ensemble structures of apo (cyan, 25 conformers, PDB: 2L28) and holo MTX-bound (raspberry, 21 conformers, PDB: 1AO8) *L. casei* DHFR illustrating the structural dynamics shift in DHFR upon MTX (green) binding as shown by the coalescence of conformers in the ensemble. RLU, relative light units. Source data are provided in a Source Data file.

measure the ligand-induced thermal stabilization of the target protein, from a purified protein or protein in a cellular setting respectively, neither isothermally assess this property[57,58]. α-complementation has been applied to CETSA facilitating in vitro target engagement detection for specific recombinantly-tagged proteins. These assays, however, require thermocycler-based instruments currently having limited throughput and accessibility[57,59]. The SDR assay does not require thermal denaturation or other treatments (*e.g.*, proteolysis) to discriminate the ligand-bound from control sample. Rather, the SDR assay is a mix-and-read type format, without necessitating separation steps, high quality antibodies, or specialized detectors, and readily scales to standard 384- and 1536-well format microtiter plates.

Ligand binding to a protein is typically characterized by a measure of $K_D$. However, in the SDR assay format ligand binding is coupled to the NLuc luminescent output where a 4-parameter nonlinear regression model is used to fit the resultant CRCs (see "Methods"). For the ligands tested in this paper with reported IC$_{50}$ and $K_D$ values, the SDR$_{50}$ mirrors the apparent affinities and rank order potency for a given compound series (*e.g.*, antifolate inhibition of DHFR, ipglycermide binding to *C. elegans* and *B. malayi* iPGMs). While the SDR assay can approximate a ligand's $K_D$ (*i.e.*, $K_D \cong$ SDR$_{50}$) for low-potency ligands, we observed some differences in the SDR$_{50}$ with, for example, *N*- versus *C*-terminal HiBiT FLuc for PTC124 in the absence of ATP (Table 1) as well as with the detection of allosteric ligands (Fig. 2d vs e). This suggests construct design can influence ligand binding, at least for low-affinity ligand binding conditions.

For the HiBiT tagged proteins in this study, basal SDR RLU varies by as much as ~35-fold (Supplementary Fig. 14), supporting a specific protein's impact on the NLuc component's accessibility or catalytic activity of reconstituted NLuc. For example, the SDR assay maximum response for FLuc-*C*-HiBiT is chemotype-dependent and varies between 20−130% among compounds described here (see Supplementary Fig. 15). For a given chemotype the maximum response may trend with the SDR$_{50}$ of the analog (Supplementary Fig. 15a), appear arbitrary (Supplementary Fig. 15b), or display cofactor-dependent synergy (*e.g.*, PTC124 plus ATP, clade A' and S chemotypes, Supplementary Fig. 15c, d).

The SDR assay format was not immediately applicable to every α-peptide fusion protein tested. The positioning of the *N*- or *C*-terminal α-peptide can influence the assay's output, an observation made particularly clear with the DNA ligases. For example, in comparing the dsDNA oligo binding to *E. coli* vs T7 DNA ligase (Fig. 4c, d), depending on the termini to which the α-peptide was appended, we observed either a significant increase or less pronounced decrease in ligand-dependent luminescence associated with DNA binding. For NAD⁺ and ATP cofactor binding, the SDR assay efficacy was consistently higher with the *N*-HiBiT ligase constructs (Fig. 4e, h), likely owing to α-peptide tag proximity to the *N*-terminal nucleotidyl transferase (NTase) subdomain (Fig. 4a). The diminished SDR output in the cofactor-treated *C*-HiBiT ligases, particularly in *E. coli* DNA ligase may be due to the flexible linker region connecting the ligase NTase and oligonucleotide binding subdomains acting as a vibrational sink[60,61].

The sensitivity of the SDR assay format to accommodate whole-cell extracts containing proteins expressed at physiological levels represents a compelling dimension of this technology. For example, from a mammalian cellular lysate containing an endogenous α-peptide tagged protein, obtained using CRISPR/Cas9-mediated homologous recombination, we recapitulated the rank order of ligand binding for DHFR antifolates compared to purified DHFR-*C*-HiBiT prepared from *E. coli* overexpression (Fig. 5d, f). This suggests that direct ligand discovery or evaluation is possible for targeting proteins expressed at endogenous cellular concentrations, more likely to form stoichiometrically relevant complexes or post-translationally modified species that might otherwise not be physiologically represented in an over-expression system.

In summary, the SDR assay technique may offer access to protein targets refractory to recombinant expression and purification, or unfeasible to assay by current methods, including those that rely on high-quality antibodies with appropriate specificity and sensitivity to the target of interest. The ability to titrate target protein-fusions in SDR to sub nM concentrations (*e.g.* Figure 3b, c) enables the sensitivity required to discriminate low nM potencies in lead optimization programs that might not be possible with competitive binding or enzyme assay formats (*e.g.*, Fig. 3d). We envision the SDR assay enabling both chemotype discovery (*e.g.*, drug and chemical probe leads such as molecular glues[62] and bifunctional agent precursors[63]) and medicinal chemistry programs for target proteins formerly incompatible with HTS[36] due to assay inefficiencies or lacking an assay platform altogether.

## Methods
### Protein expression plasmid constructs
All constructs were prepared and sequenced by either GenScript Corporation (Piscataway, NJ, USA; www.genscript.com) or by Bio Basic (Amherst, NY, USA; www.biobasic.com). Bacterial expression constructs were assembled in the pET21a vector. Constructs designated with '-HiBiT' contain the canonical NLuc α-complementation 11 amino acid peptide sequence, VSGWRLFKKIS[5].

**Plasmid deposition.** All plasmids were deposited and available from Addgene (www.addgene.org) where sequences were verified. The plasmids have the following Addgene IDs: FLuc-*N*-HiBiT (ID 207117), FLuc-*C*-HiBiT (ID 207118), FLuc-*C*-NLuc (ID 234566), human (Hs) DHFR-*C*-HiBiT (ID 207119), Hs ABL1-*N*-HiBiT kinase domain (ID 207121), ABL1-*C*-HiBiT (ID 234568), Hs PKA-*N*-HiBiT (ID 211377), *C. elegans* iPGM-*C*-HiBiT, (ID 207120), *B. malayi* iPGM-*C*-HiBiT(ID 218483), *E. coli* Lig-*N*-HiBiT (ID 215437), *E. coli* Lig-*C*-HiBiT (ID 215438), T7 Lig-*N*-HiBiT (ID 215479), T7 Lig-*C*-HiBiT (ID 215441), and NLuc (ID 234567).

**Bacterial expression constructs.** *E. coli* expression plasmid pET21a was used to prepare proteins containing either *N*- or *C*-terminal HiBiT sequence. Each construct contains a His tag separated by a thrombin-sensitive sequence. The 33 bp HiBiT sequence used was GTGAGCGGCTGGCGGCTGTTCAAGAAGATTAGC[5].

**FLuc-C-NLuc (FLuc-NLuc-His10), FLuc-N-HiBiT (HiBiT-FLuc-His10) and FLuc-C-HiBiT (FLuc-HiBiT-His10).** The modified *Photinus pyralis* (common eastern firefly) luciferase (GenBank ID: M15077.1) cDNA was synthesized and cloned using the BamHI and XhoI restriction sites, containing the protein sequence Glu2-Val550. *FLuc-NLuc-His10* was prepared following a similar protocol where the 11 aa HiBiT peptide was replaced with the 19.1 kDa NanoLuc luciferase protein.

**ABL1-N-HiBiT (His10-HiBiT-ABL1) and ABL1-C-HiBiT (ABL1-HiBiT-His10).** The human Abelson tyrosine kinase (ABL1) domain (NCBI NG_012034.1 REGION: 4163.178795) cDNA, containing amino acid residues Asp233-Ser500, was synthesized and cloned using BamHI and XhoI restriction sites.

**PKA-N-HiBiT (His10-HiBiT-PKA).** The modified human serine/threonine protein kinase cAMP-activated catalytic subunit alpha (GenBank: BC108259.1; NCBI Gene ID 5566) cDNA, containing amino acids Gly2-Phe351, was synthesized and cloned into the BamHI and NotI restriction sites.

**C. elegans iPGM-C-HiBiT (C. elegans iPGM-HiBiT-His6).** The HiBiT sequence was subcloned into the *Hin*dIII restriction site of *C. elegans* iPGM-6xHis, described in Yu H et al., between the *C. elegans* cofactor-independent phosphoglycerate mutase (NCBI Reference Sequence NP_491896.1) cDNA, containing Phe2-Ile539 and the 6xHis tag[41].

**B. malayi iPGM-C-HiBiT (B. malayi iPGM-HiBiT-His6).** The HiBiT sequence was subcloned into the BamHI restriction site of *B. malayi* iPGM-6xHis, described in Yu H et al., between the *B. malayi* cofactor-independent phosphoglycerate mutase (NCBI Gene ID AAQ97626.1) cDNA, containing Ala2-Glu515 and the 6xHis tag[41].

**E. coli DNA Lig-N-HiBiT (His10-HiBiT-E. coli Lig) and E. coli Lig-C-HiBiT (E. coli Lig-HiBiT-His10).** The modified NAD-dependent DNA ligase LigA (NCBI Ref. Seq. WP_063099908.1) cDNA was synthesized and cloned using the EcoR1 and XhoI restriction sites, containing the protein sequence Glu2-Ser671.

**T7 DNA Lig-N-HiBiT (His10-HiBiT-T7 Lig) and T7 Lig-C-HiBiT (T7 Lig-HiBiT-His10).** The modified *Escherichia phage* T7 DNA Ligase (Gene ID 1261055) cDNA was synthesized and cloned using the BamHI and XhoI restriction sites, containing the protein sequence Met2-Met359.

**DHFR-C-HiBiT (DHFR-HiBiT-His10).** The modified human dihydrofolate reductase (NCBI Gene ID 1719) cDNA was originally synthesized and cloned using BamHI and *Hin*dIII restriction sites, containing the protein sequence Val2-Asp187 and a *C*-terminal Avi tag[64]. Using this plasmid, a HiBiT-AAG tag was synthesized and subcloned into *DHFR-C-AviTag-His10* before the 10xHis, replacing the existing Avi tag.

### Kinase substrates, Ipglycermide cyclic peptides and Ce-2d-FL FP probe
**Reagents.** Fmoc amino acids, resins, and coupling reagents were obtained from commercial sources (CEM Corp., Matthews, North Carolina; Chem-Impex, Wood Dale, Illinois; Sigma-Aldrich, St. Louis, Missouri; and ChemPep, Wellington, Florida), and used without further purification. 6-Carboxyfluorescein N-hydroxysuccinimide ester (FAM NHS ester, 6-isomer), was from BroadPharm (Cat # BP-23900, CAS # 890090-41-2).

**Kinase peptide substrates.** Abltide (H-Lys-Lys-Gly-Glu-Ala-Ile-Tyr-Ala-Ala-Pro-Phe-Ala-NH$_2$) and kemptide (H-Leu-Arg-Arg-Ala-Ser-Leu-Gly-NH$_2$) were synthesized (100 µmol scale) by standard Fmoc solid phase peptide synthesis on a Liberty Blue HT12 automated peptide synthesizer (CEM Corp.) using previously published[41,65,66] and standard manufacturer's procedures[67]. *C*-terminal amide peptides were synthesized on Novabiochem Rink Amide AM resin. Amide coupling used N, N'-diisopropylcarbodiimide (DIC) and ethyl (hydroxyamino)cyanoacetate (oxyma) in DMF. Fmoc-protecting groups were removed using 20% piperidine supplemented with 0.1 M oxyma in DMF. The peptides were purified using preparative HPLC and characterized by LC-MS methods below. LC-MS (abltide) $t_r$ = 3.34 min; HRMS (ESI) m/z (M + H)$^+$ calcd for C$_{60}$H$_{93}$N$_{15}$O$_{15}$, 1264.70; found, 1264.73. (M + 2H)$^{2+}$ calcd for C$_{60}$H$_{93}$N$_{15}$O$_{15}$, 632.86; found, 632.87. (M + 3H)$^{3+}$ calcd for C$_{60}$H$_{93}$N$_{15}$O$_{15}$, 422.24; found, 422.25. Purity (A214): 97.55%. LC-MS (kemptide) $t_r$ = 1.15 min; HRMS (ESI) m/z (M + H)$^+$ calcd for C$_{32}$H$_{62}$N$_{14}$O$_8$, 771.49, 772.50; found, 771.49, 772.49. (M + 2H)$^{2+}$ calcd for C$_{32}$H$_{62}$N$_{14}$O$_8$, 386.25, 386.75; found, 386.25, 386.75. Purity (A214): 88.66%. Kemptide is also commercially available from several sources,

including Promega (Cat #V5601), Anaspec (Cat#AS-22594), and Thermo Scientific (Cat# J60591-#0). Abltide is also commercially available from several sources, including Anaspec (Cat#AS-61749) and Sigma-Aldrich (Cat#12-493).

**Ipglycermide cyclic peptides.** The following cyclic peptides used in this study, Ce-2, Ce-2d, Ce-2d Y11K, Ce 2-P4V, Ce 2-Y7F, Ce-2d P4V, Ce-2d Y7W, Ce-2d Y7-(4 f)F, Ce-2d Y7H, and Ce-2d Y7f were prepared using automated solid phase peptide synthesis[16].

**Ce-2d-FL.** The fluorescein-labeled Ce-2d fluorescence polarization (FP) assay probe, Ce-2d-FL, was prepared by acylation of Ce-2d Y11K. To Ce-2d Y11K (0.01 g, 6.95 μmol, 1eq) in dimethyl formamide (250 μL), was added N, N-diisopropylethylamine (24 μL, 20 eq) followed by FAM NHS ester, 6-isomer (6.58 mg, 0.014 mmol, 2 eq). The reaction mixture was stirred at room temperature for 1 h then diluted with 0.5 mL methanol. The product was purified using preparative HPLC and characterized by LC-MS methods below. LC-MS (Ce-2d-FL) $t_r = 3.92$ min; HRMS (ESI) m/z (M + H)$^+$ calcd for $C_{89}H_{98}N_{13}O_{26}S$, 1796.65; found, 1796.65.

**Preparative HPLC.** Preparative purification was performed on a Waters semi-preparative HPLC using an Agilent ZORBAX XDB-C18 column of 5-micron, 30 ×100 mm size at a flow rate of 45 mL/min. The mobile phase consisted of acetonitrile and water each containing 0.1% trifluoroacetic acid. A gradient of 10% to 50% acetonitrile over 8 min was used during the purification. Fraction collection was triggered by mass detection. The product was pooled and lyophilized to obtain the pure ( ≥ 95%) powder.

**HRMS analysis.** Mass was confirmed based on HRMS as determined on a HPLC-TOF system (Agilent 6210 TOF system equipped with an Agilent 1200 HPLC system). Samples (10 to 100 μM) were resolved on a reversed phase analytical column (Extend-C18, 3.5 μm, Agilent Technologies) eluted using 4-100% aqueous ACN (0.1% formic acid). MS spectra were acquired from m/z 350 to 1700 or 400 to 3000 at a scan rate of 1 spectrum per second with Profile format. The electrospray ionization (ESI) source parameters were used as follows: gas temperature, 340 °C; gas flow, 10 L/min; nebulizer, 50 psi; fragmentor, 175 V.

### Expression and purification of proteins from *E. coli*

Proteins were prepared according to previous methods with indicated modifications[16,41,64,68]. In general, the expression plasmid was transformed into BL21(DE3) *E. coli* competent cells and 50 μL of cell mixture was plated on an LB agar plate containing 50 μg/mL ampicillin. A starter culture (50 mL LB with 50 μg/mL ampicillin, or LB/AMP50) was inoculated with a colony streak and the culture was shaken overnight at 37 °C. Four 1 L LB/AMP50 in 2800 mL flasks were each inoculated with 10 mL overnight starter culture and were shaken at 200 rpm at either 30 or 37 °C until $OD_{600} = 0.4-0.5$. The cultures were cooled to 16 °C for 15 min, then induced with IPTG (0.5-1.0 mM, final concentration) and allowed to shake overnight at 200 rpm and 16 °C. Cells were then harvested by centrifugation for 10 min at 7000 rpm and 4 °C, and the resulting cell pellet was stored at −80 °C until purification.

Frozen cell pellets were resuspended in 30 mL Buffer A (20 mM sodium phosphate, pH 7.4, 300 mM sodium chloride, 10 mM imidazole, 10% glycerol) and one EDTA-free protease inhibitor cocktail tablet (Millipore Sigma, Cat # 5056489001) was added. The cells were lysed by six sonication cycles (30 s on, 60 s off). The resulting lysate was centrifuged for 25 min at 20000 rpm and 4 °C to separate the insoluble cell debris from soluble proteins.

**Ni-NTA Agarose affinity chromatography.** All target proteins were initially enriched from *E. coli* lysates by immobilized metal ion affinity chromatography. Briefly, the post-centrifugation lysis supernatant was loaded onto a preequilibrated His Trap FF prepacked column (with 5 mL bed volume) at 1.5 mL/min flow rate and washed with 5 column volumes of Buffer A and 5 column volumes of 92% Buffer A and 8% Buffer B (20 mM sodium phosphate, pH 7.4, 300 mM sodium chloride, 500 mM imidazole, 10% glycerol). The target protein was eluted with a linear gradient of 10–100% Buffer B (50–500 mM imidazole). Protein fractions containing the target protein were pooled and concentrated by Amicon centrifugal concentrators (10 KDa MWCO, Cat # UFC901024).

**Secondary column chromatography.** Purity was assessed by denatured SDS-PAGE, protein targets less than 80% pure were further isolated by size exclusion or anion exchange chromatography, see SDS-PAGE analysis of recombinantly *E. coli* expressed and purified proteins (Supplementary Fig. 16). The pooled protein fractions were concentrated to less than 2 mL volume and loaded onto a Hi-Load Sephadex 16/60 prepacked column (bed volume of 120 mL) (Cytiva) preequilibrated with 1.5 column volumes target-specific storage buffer (FLuc, DHFR, iPGMs: 150 mM Tris-HCl, pH 8.0, 25 mM magnesium sulfate, 100 mM potassium chloride, 10% glycerol; kinases: 20 mM Tris-HCl, pH 7.3, 100 mM sodium chloride, 5 mM magnesium chloride, 10% glycerol; ligases: 10 mM Tris-HCl, pH 7.4, 50 mM potassium chloride, 1 mM dithiothreitol, 0.1 mM EDTA). Proteins were resolved by size exclusion at a flow rate of 0.5 mL/min. Relative purity of protein fractions were determined by denaturing 4-12% or 4-20% SDS-PAGE, and fractions containing protein of correct molecular weight were pooled and concentrated by Amicon centrifugal concentrator. If the target protein required further purification, the concentrated protein was injected over a HiTrap Q HP (Cytiva) anion exchange column and resolved over a linear buffer gradient up to 0.5 M NaCl in PBS at 1.5 mL/min. The pure protein concentration was measured by Nanodrop $A_{280}$ absorbance reading and by BCA assay (Thermo Scientific, cat # 23225) and final purity was assessed with SDS-PAGE (Supplementary Fig. 16). Final stocks were stored in 50 μL aliquots containing 20% glycerol and flash frozen for cryopreservation at -80 °C unless otherwise stated. Ligases were preserved with 50% glycerol and stored at 4 °C during testing.

**FLuc-N-HiBiT, FLuc-C-HiBiT, FLuc-C-NLuc, and NanoLuc.** Four x 1 L LB cultures grown in 2.8 L baffled Fernbach culture flasks containing 50 μg/mL ampicillin were shaken at 200 rpm and 37 °C until $OD_{600} = 0.4-0.5$, induced by adding IPTG to a concentration of 1 mM, and grown overnight at 16 °C with continued 200 rpm shaking. Ni-NTA purification of FLuc-*N/C*-HiBiT eluted as a mostly pure protein of size -65.9 kDa, while FLuc-*C*-NLuc eluted as a pure protein at -83 kDa. The target protein was further purified by size exclusion chromatography (SEC) as described above to yield > 90% pure FLuc-*N*-HiBiT, FLuc-*C*-HiBiT, and FLuc-*C*-NLuc products. Final aliquots were prepared for cryopreservation as described above. FLuc-*N*-HiBiT, FLuc-*C*-HiBiT, and FLuc-*C*-NLuc appear stable at 4 °C in 20% glycerol for up to 3 weeks. Nanoluciferase protein was prepared following a similar expression and purification method to yield a highly pure protein of -22.7 kDa in size.

**ABL1-N-HiBiT and ABL1-C-HiBiT kinase domain.** Eight x 1 L LB cultures supplemented with 50 μg/mL ampicillin were shaken at 200 rpm and 37 °C until $OD_{600} = 0.4-0.5$, cooled briefly for 15 min, induced by adding IPTG to a concentration 1 mM from freshly prepared liquid stock, and then grown overnight at 16 °C with shaking at 200 rpm. ABL1-*N*-HiBiT expression was relatively weak and only partially pure following Ni-NTA chromatography. Subsequent size exclusion chromatography was able to resolve the target protein from contaminants to yield -540 μg of > 80 % pure ABL1-*N*-HiBiT (36.9 kDa). Protein was aliquoted in storage buffer containing 20% glycerol and flash frozen

for storage at −80 °C. In our experience, ABL1-*N*-HiBiT and ABL1-*C*-HiBiT are unstable at 4 °C and should be used immediately after thawing from −80 °C aliquot.

**PKA-N-HiBiT**. 100 mL of LB culture supplemented with 50 µg/mL of ampicillin was inoculated by single bacterial colony transfer and grown to OD$_{600}$ = 0.4 − 0.5 at 37 °C with shaking. Cultures were then cooled for 15 min, induced with 1 mM IPTG, and incubated overnight at 16 °C. Total bacterial lysate was cleared by centrifugation at 15,000 g for 30 min, and then the soluble lysate fraction was loaded onto a 100 µL Ni-NTA His SpinTrap column (Cytiva 28401353) preequilibrated with buffer A. Column was sequentially washed with 400 µL Wash 1 buffer (85:15 Buffer A:B) and 400 µL Wash 2 buffer (70:30 buffer A:B) using 100 g centrifugation. Target protein was eluted by addition of 150 µL buffer B, and elution fractions were supplemented with addition glycerol for cryopreservation. Eluted protein was assessed by SDS-PAGE for yield and purity. PKA-*N*-HiBiT was recovered in moderate yield as a > 95% pure single band at 45.5 kDa. Protein has remained stable through multiple snap freeze/thaws when stored at −30 °C in 20% glycerol.

**C. elegans and B. malayi iPGM-C-HiBiT**. The nematode iPGM-HiBiT-His6 constructs were expressed and purified according to previously published methods[16,41]. Briefly, following Ni-NTA chromatography, and size-exclusion chromatography, protein purity was determined by denaturing 4-12% SDS-PAGE to be > 95% (*C. elegans* and *B. malayi* iPGM expected MW 65.0 kDa and 61.0 kDa, respectively), concentration determined, supplemented with glycerol (20%), and aliquoted into 20 µL fractions for -80 °C cryopreservation. Prior to use diluted enzyme (1 mL) was supplemented with 10 µL of 10 mM ZnCl$_2$ and 10 µL of 10 mM MnCl$_2$ then dialyzed for 16 h (4 °C) against 1 L of 1X storage buffer containing 20% glycerol using Slide-A-Lyzer™ Dialysis Cassettes (Thermo Scientific, Cat # 66380). The dialyzed samples were stored at 4 °C up to 8 weeks.

**DNA ligases**. The DNA ligase-HiBiT constructs (*E. coli-N*, *E. coli-C*, T7-*N*, T7-*C*) were expressed and purified as described for PKA-*N*-HiBiT. The eluted proteins were assessed by SDS-PAGE for yield and purity. Each DNA ligase-HiBiT construct was isolated in high yield as a single, > 95% pure monomeric band corresponding to the following molecular weights: *E. coli*-HiBiT (78.6 kDa), T7-HiBiT (45.9 kDa). Enzymes were further purified and buffer exchanged into ligase storage buffer (see above) by SEC. Initial investigation suggests ligase enzymes may lose fidelity following freezing. Proteins were stored at 4 °C following SEC in storage buffer supplemented with 50% glycerol and 200 µg/ml BSA.

**DHFR-C-HiBiT**. The human DHFR-*C*-HiBiT construct was expressed and purified according to previously published methods[64]. Briefly, following Ni-NTA chromatography, and SEC, protein purity was determined by denaturing 4−12% SDS-PAGE to be > 95% (expected MW 25.7 kDa), concentration determined, and supplemented with glycerol (20%) for cryoprotection. Protein solution was aliquoted into 20 µL fractions, flash frozen in liquid nitrogen, and stored at −80 °C until further use. Proteins appear stable at 4 °C in 20% glycerol for up to 3 weeks.

## Genome edited mammalian DHFR-*C*-HiBiT cell line and lysate preparation

A HEK293 cell line constitutively expressing *S. pyogenes* Cas9 (Gene-Copoeia, Cat # SL553) was used to generate the *C*-terminal edit of the DHFR gene (DHFR-*C*-HiBiT) applying general methodology previously described[24,52,53]. The DHFR 3′-gene sequence surrounding the stop codon (underlined), aggagaaaggcattaagtacaaatttgaagtatatgagaagaatg attaatatgaaggtgtttt ctagtttaagttgttcccctccctctgaaaaaagt, used to generate the CRISPR-Cas9 crRNA sequence, GAGAAGAAUGAUUAA

UAUGA, was accomplished with IDT software (Custom Alt-R CRISPR-Cas9 guide RNA). The gRNA was prepared by annealing the crRNA with Alt-R CRISPR-Cas9 tracrRNA (IDT Technologies, 1072533).

The DHFR/HiBiT 3′-donor DNA used for homology directed repair (HDR), purchased as an Alt-R HDR single-stranded Ultramar DNA Oligonucleotide (IDT), contains 70 bp homology arms (lower case sequences) flanking the 33 bp HiBiT sequence (upper case sequence) followed by a stop codon (underlined):

HDR cassette: tgttctctctgatgtccaggaggagaaaggcattaagtacaaatttg aagtatatgagaagaatgatGTGAGC

GGCTGGCGGCTGTTCAAGAAGATTAGCtaatatgaaggtgttttctagttt aagttgttcccctccctctgaaaaaagtatgtgtattttttacattagaaa

**HEK293 cell line electroporation**. Briefly, using a 4D Nucleofector System (Lonza) and program CM-130, the Cas-9 expressing HEK293 cells (5E10$^5$) were electroporated with 120 pmol gRNA and donor DNA (100 pmol) resuspended in 20 µL of 4D Nucleofector solution S for HEK293 cells (Lonza). Cells were incubated at ambient temperature before transferring to the DMEM, supplemented with 10% FBS and 1% Penicillin-Streptomycin (P/S) growth media. Transfection efficiency was estimated using the control transfection reaction with GFP plasmid after 24 and 48 h transfection. After recovery and confirmation of HiBiT signal, cells were dispensed for monoclonal expansion into three 96 well plates using a SH800S cell sorter (Sony Biotechnology).

**Cell lysate preparation and characterization**. For lysate preparation, the DHFR-*C*-HiBiT gene edited HEK293 monoclonal cell line was grown in a T75 flask (DMEM, 10% FBS, 1% P/S) to confluence. Media was removed, cells gently rinsed with warm 1X PBS, incubated for 3 min with 3 mL PBS containing 0.25% trypsin, then collected with 5 mL warm growth media, pelleted and rinsed 3x with PBS. The cell pellet was lysed by adding 500 µL ice-cold NP-40 lysis buffer (ThermoFisher Scientific, Cat # FNN0021) containing 5 µL 100x protease inhibitor cocktail in DMSO (Sigma Aldrich, Cat # P8340-1mL) incubated on a rocker at 4 °C for 1 h (protected from light), centrifuged at 15,000 g for 10 min at 4 °C, after which supernatant was transferred to a 1.5 mL Eppendorf tube, and frozen at −80 °C. A sample of the lysate was used for the estimation of DHFR-*C*-HiBiT concentration by Western blot with anti-DHFR antibody (Abcam, Cat # ab124814; Clone# EPR5285; RRID:AB_10975115) and NanoBlot (Promega, Cat # N2410) analysis as described in Supplementary Fig. 13.

## Thermal shift profiles (Thermofluor assay)

Thermal stabilization of proteins in the presence of the ligands were determined using Thermofluor technique[69]. MicroAmp optical 384-well plates (Applied Biosystems by Life Technologies), ViiA 7 real-time PCR instrument (Applied Biosystems by Life Technologies) and SYPRO Orange (Life Technologies) were used to test the samples. The final concentration of the protein, compounds and dye were maintained at 5 µM, 100 µM and 10x respectively in a 20 µL reaction volume. The proteins and compounds were diluted in PBS (10 mM Na$_2$HPO$_4$, 1.8 mM KH$_2$PO$_4$, 137 mM NaCl, and 2.7 mM KCl). Proteins were added to the 384-well plate first followed by the compounds. Samples were incubated for 10 mins at room temperature and protected from light. SYPRO Orange dye was diluted to 50x from 5000x stock in a PBS buffer, and 4 µL dye was added to the each well using a multi-channel pipette. The plate was sealed with a MicroAmp optical adhesive film, centrifuged at 1500 RPM for 2 min, and then loaded onto the ViiA 7 real−time PCR instrument for fluorescence intensity measurements upon heating the sample from 25 to 95 °C at 1 °C/min increments. Data was exported to excel, fluorescence intensity data and its first derivative were normalized and plotted using GraphPad Prism to determine $T_m$. See Supplementary Table 3 for additional details.

## Functional enzyme and competition-binding assays

**Firefly luciferase (FLuc) enzymatic luminescence assay.** The enzyme activity of FLuc-*N*-HiBiT, FLuc-*C*-HiBiT, and FLuc-*C*-NLuc was measured similar to *P. pyralis* firefly luciferase (FLuc) as previously described[70]. Briefly, 1–10 nM purified FLuc-*N*-HiBiT, FLuc-*C*-HiBiT, or FLuc-*C*-NLuc in FLuc ATP Assay Buffer (66.6 mM Tris Acetate; pH 7.6, 10 mM Magnesium Acetate, 0.01% Tween-20, 0.05% BSA, 10 μM ATP) was dispensed at 4 μL/well into a 1536-well Greiner Bio-One white, high solid base, medium bind plate (Cat # 789175-F) using a BioRaptr 2 (LGR, Inc.). Compounds and controls were added to each plate as described in Supplementary Data 8 with a Hornet pintool system (Wako Automation). Compounds were incubated with respective enzymes for 30–60 min at room temperature protected from light and covered by solid metal lid to prevent evaporation. A 23 μM solution of D-luciferin was prepared in 1X PBS, filtered, and then 3 μL of luciferase substrate was delivered to each well using a BioRaptr 2 dispenser, for a final concentration of 10 μM D-luciferin (Sigma Aldrich; L6152). Luciferin was incubated with FLuc enzyme for 5 min before FLuc activity was then measured on a ViewLux CCD imager (PerkinElmer), with exposure times up to ten seconds with medium to high gain settings. Data were normalized to average no enzyme control as 100% enzyme inhibition and average DMSO neutral control. See Supplementary Table 4 for additional details.

**NanoLuc luciferase (NLuc) enzymatic luminescence assay.** NLuc enzyme activity was measured similar to that described for FLuc enzymes, where 1 nM NLuc was plated, treated and incubated as above. Compounds and controls were added to plates as described in Supplementary Data 8. Nano-Glo luciferase assay reagent (Promega) was prepared fresh according to manufacturer's protocol, filtered and added at 3 μL per well across assay plates with a BioRaptr2 dispenser. Enzyme was incubated with Nano-Glo reagent for 10 min and NLuc luminescence was read on a ViewLux CCD imager (PerkinElmer), with 1 s exposure with medium gain and 2x binning. Data were normalized to average no enzyme control as 100% enzyme inhibition and average DMSO neutral control. See Supplementary Table 4 for additional details.

**Abelson Kinase (ABL1) and Protein Kinase A (PKA) peptide phosphorylation.** Kinase activity was measured by ATP depletion using the Kinase-Glo Plus (KGP) assay kit (Promega) in 1536-well plate format[71]. Abltide, H-KKGEAIYAAPFA-NH$_2$ (ABL1 kinase substrate), and kemptide, H-LRRASLG-NH$_2$ (PKA substrate) were prepared using a Liberty Blue HT12 automated peptide synthesizer (CEM Corp.) as described above. ABL1-*N* or -*C*-HiBiT and PKA-*N*-HiBiT were prepared in kinase assay buffer (final concentrations: 17.5 nM kinase, 20 mM Tris-HCl, pH 7.2, 5 mM magnesium chloride, 150 mM potassium chloride, 0.05% IGEPAL, 2 mM dithiothreitol, 87.5 μM ATP) and dispensed at 4 μL/well into columns 2–48 of white, solid bottom medium bind 1536-well microtiter plates using a BioRaptr 2. Controls were added with a Mosquito dispenser (STP LabTech) while compounds were transferred with a pintool to each plate as described in Supplementary Data 8 and incubated for 30 min at ambient temperature. Peptide substrates were added to assay plates at a final concentration of 62.5 μM by Mosquito dispenser and incubated with kinase enzyme for 3 h at room temperature, covered and protected from light. Kinase activity was determined by addition of 4 μL/well of KGP reagent using a BioRaptr 2. Following a 10 min incubation, KGP luminescence was measured with a ViewLux plate reader with 1 sec exposure, with medium to high gain setting, slow speed and 2x binning. Data were normalized to the average of no enzyme control column as 100% kinase inhibition and DMSO control column as 100% phosphorylation reaction progress. See Supplementary Table 5 for additional details.

**Co-factor independent phosphoglycerate mutase (iPGM) assay.** The enzyme activity of *C. elegans* or *B. malayi* iPGM-*C*-HiBiT was measured using the functional coupled-enzyme luminescence endpoint output assay for iPGM as previously described[16,41,68]. Briefly, the indicated concentration of 4 μL of iPGM-*C*-HiBiT was dispensed into 1536-well white, high solid base, medium bind plates in a pH 8.0 iPGM assay buffer (30 mM Tris-HCl, 5 mM magnesium sulfate, 20 mM potassium chloride and 0.08% BSA) with the BioRaptr 2. Compounds and controls were added to each plate with a pintool as described in Supplementary Data 8. Enzymes were incubated with peptide for > 20 min at ambient temperature, protected from light. Two microliters of 3PG-containing coupled-enzyme assay buffer (including 0.2 U enolase, 0.3 U pyruvate kinase and 3 mM ADP) was added to enzyme/compound mixtures with a BioRaptr 2 and reactions were incubated for 5 min at ambient temperature, followed by addition of 4 μL KGP reagent. Enzyme was at 5 nM final concentration for the standard assay with an estimated $K_M$ 3-phosphoglycerate (3PG) substrate concentration of 400 μM. Plates were incubated at ambient temperature for 10 min, protected from light, then measured by a ViewLux plate reader. Data was row-wise normalized to enzyme with DMSO as neutral control and without enzyme with vehicle control as 100% inhibition. See Supplementary Table 6 for additional details.

**C. elegans iPGM fluorescent polarization competition-binding assay.** All reagents were prepared on ice, protected from light. *C. elegans* iPGM-*C*-HiBiT enzyme was dispensed at 4 μL/well into 1536-well black, solid-bottom microplates (Greiner Bio-One, Cat # 789176-F) with a BioRaptr 2 in iPGM assay buffer containing 0.05% IGEPAL (v/v) at a final concentration of 10 nM of enzyme. Compounds and controls were added with a pintool to each plate as described in Supplementary Data 8. Assay plates were incubated for 20 min at ambient temperature. A pre-fluorescent ligand fluorescence polarization measurement was read on the Spark multimode microplate reader (Tecan) with the following optical settings: monochromator excitation wavelength 480 nm (20 nm bandwidth filter), emission wavelength 535 nm (20 nm bandwidth filter), 30 flashes/well and manual settings of G-factor 1.0, Z-position 20000, optimal gain, and automatic mirror. Then 2 μL of fluorescein-labeled macrocyclic Ce-2d-FL in iPGM assay buffer was dispensed into each microplate well via BioRaptr 2 for a final assay concentration of 5 nM probe. Microtiter plates were allowed to reach equilibrium (30 min incubation at ambient temperature) while protected from light. Fluorescence polarization was measured with the same reader settings as above. Data were row-wise normalized to no enzyme control as -100% activity and enzyme + DMSO vehicle control as neutral control. See Supplementary Table 7 for additional details.

**Ligase-HiBiT DNA ligase activity agarose gel assay.** Ligase reactions were prepared using optimized concentrations of Ligase-HiBiT enzyme determined qualitatively through replicate gel assays: 1.0 μM *E. coli* Lig-*N*-HiBiT, 100 nM *E. coli* Lig-*C*-HiBiT, 100 nM T7 Lig-*N*-HiBiT, 1.0 μM T7 Lig-*C*-HiBiT. Control *E. coli* and T7 ligase were purchased from New England Biolabs and prepared according to manufacturer protocol using commercial buffers. Ligase reactions (30 μL) were prepared for *E. coli* Lig-*N/C*-HiBiT in *E. coli* Ligase buffer (NEB) and T7 Lig-*N/C*-HiBiT in StickTogether DNA Ligase buffer (NEB). Lambda/HindIII DNA substrate working stock prepared in 10 mM Tris buffer, pH 8.0, and was added to each ligase reaction such that loading 30 μL of the loading-dye quenched reaction mixture onto the gel equaled 200 ng total DNA per well. Ligation reactions were incubated for 60 min at ambient temperature protected from light. Reactions were heat inactivated for 20 min at 65 °C, chilled briefly on ice, then centrifuged at 12,000 g for five minutes. Ligation progress was qualitatively assessed following resolution over 1% agarose gel electrophoresis at 75 V for 90 min and imaged using GreenGlo Safe

fluorescent dye (Denville Scientific) with an iBright gel imager (ThermoFisher Scientific). See Supplementary Table 8 for additional details.

**Dihydrofolate reductase enzymatic absorbance assay.** To accommodate consistent compound addition used in this study, DHFR enzymatic activity[72,73] was measured in a 1536-well miniaturized assay adapted from the Sigma-Aldrich dihydrofolate reductase assay (Cat # CS0340). In this format 4 μL/well DHFR-C-HiBiT was dispensed in 1536-well black, clear-bottom, low base microplates (Aurora Discovery, Cat # 00019324) via a BioRaptr 2 for a final concentration of 100 nM in 1X assay kit buffer (Sigma-Aldrich, Cat # A5603) containing 75 μM NADPH (Sigma-Aldrich, Cat # N6505). Compounds and controls were added as above to each plate as described in Supplementary Data 8. DHFR-C-HiBiT was incubated with compounds for 20 min at ambient temperature. Background absorbance of compounds was measured for each assay plate prior to substrate addition on the Spark multimode microplate reader at 340 nm with a bandwidth of 5 nm and 25 flashes/well. Then 2 μL of dihydrofolate (DHF, Sigma-Aldrich, Cat # D7006) substrate, freshly prepared according to manufacturer's protocol and diluted in 1X assay kit buffer, was dispensed into each microplate well with a BioRaptr 2 for a final assay DHF concentration of 100 μM. DHFR-C-HiBiT was incubated with substrate for 10 min at ambient temperature, protected from light and NAPDH absorbance (340 nm) was measured on a Spark multimodal microplate reader as above. Inhibition of enzyme activity resulted in an absorbance increase; therefore, data was normalized to enzyme in the presence of DMSO vehicle control as neutral control and DMSO treated buffer without enzyme as 100% activity. See Supplementary Table 9 for additional details.

**Structural Dynamics Response (SDR) assays**
SDR assays were measured as luminescence output from NanoLuc luciferase (NLuc) or reconstitution of active enzyme by α-complementation of the target protein N- or C-terminal HiBiT peptide with added LgBiT protein, and NLuc substrate. Protein-HiBiT targets were expressed and purified to various levels of homogeneity and optimized at indicated concentrations, not necessarily the lowest possible concentrations, to produce consistent luminescence with statistically significant signal above no protein background control in target specific assay buffer (see Supplementary Table 11). Four μL/well buffer control (column 1) and NLuc or HiBiT tagged proteins (columns 2-48) were dispensed into 1536-well Greiner Bio-One white, high solid base, medium bind plates with a BioRaptr 2. Compounds and controls were added to the microtiter plates as described in Supplementary Data 8, then incubated and protected from light for 30 min at ambient temperature. A modified dispensing protocol was developed for aqueous ligands that were unable to be transferred by pintool (see Supplementary Table 10 aqueous ligand dispense section). The substrate solution was prepared according to manufacturer's protocol where ω-fragment (LgBiT) was added at 1:100 dilution and NLuc substrate (furimazine) was prepared at 1:50 dilution in Nano-Glo HiBiT lytic buffer (Promega, Cat # N3040). This substrate solution, 3 μL/well, was added with a BioRaptr 2 and assay plates were protected from light and incubated for 10 min at ambient temperature. NLuc luminescence was measured on a ViewLux CCD plate reader with optimized reader settings to each protein target (see Supplementary Tables 10, 11 and Supplementary Fig. 14 for additional details).

**Compound libraries**
**Firefly luciferase (FLuc) inhibitor library.** The firefly luciferase (FLuc) inhibitor library was designed based on available PubChem data from FLuc chemical library profiling (PubChem AIDs 2309) and literature on FLuc inhibitor chemotypes[25,74] with the aim of encompassing chemotype analogs having a range of FLuc inhibitory activity and structural diversity. Briefly, chemotype cores identified to accompany FLuc inhibitors were used to search the NCATS compound repository for analog series. A total of 1,343 compounds (Supplementary Data 1) were acquired and formatted into an 11-point 1536-well inter-plate titration series[75]. This 11-plate compound archive contained a highest stock concentration of 10 mM compound (in DMSO) where each subsequent plate contained a 1:3 dilution in DMSO.

**FLuc clade S analog library.** The clade S core structure was used to search the internal NCATS compound library collection for analogs with a Tanimoto similarity ≥0.9. 88 additional compounds were identified and arrayed with the original 48 clade S molecules and representatives from across the FLuc inhibitor library (Supplementary Data 3). Compounds were plated across two 1536-well plates in 11-pt, 1:3 titration in an intra-plate format with a high concentration of 10 mM (in DMSO).

**Protein kinase inhibitor sub-library.** From Pharos[76] query target-annotated list of 1177 kinase inhibitors, 51 including ABL1 as a primary or secondary target were selected. An additional 77 kinase inhibitors were selected from the remaining 1126 compounds randomly such that no two compounds had greater than 0.5 Tanimoto similarity. The selected kinase inhibitors were used to create a single 1536-well plate containing 128 compounds (Supplementary Data 4) as 1:3 intra-plate dilutions where the highest concentration was 10 mM (in DMSO) and the lowest, 0.17 μM (in DMSO).

**Data analysis**
**Assay statistical analyses.** Assay statistics including coefficient of variation (CV, Supplementary Equation (1)), signal-to-background (S:B, Supplementary Equation (2)), signal-to-noise (S:N, Supplementary Equation (3)), and Z' (Supplementary Equation (4)) were calculated as previously described[27,77] using the controls listed in Supplementary Data 7.

**Curve fitting analysis.** Concentration response curves (CRCs) for inhibitors or ligands to specific targets were fit and $IC_{50}$ or $SDR_{50}$ values were initially calculated using the standard Hill equation, e.g., nonlinear regression log(compound) vs. response -- variable slope (four parameters) fit (Supplementary Equation (5)) or bell fit (Supplementary Equation (6)) in Prism 8 (GraphPad Software), and constraints added until a fit approximating the data points was achieved. All $IC_{50}$ and $SDR_{50}$ values are stated as $pIC_{50}$ (Supplementary Equation (7)) and $pSDR_{50}$ (Supplementary Equation (8)) after conversion to molar concentration in the main text. All $IC_{50}$ values are apparent $IC_{50}$ values (i.e., as determined under the reported conditions).

**Quantitative HTS data analysis.** qHTS data was normalized to the averages of the respective intra-plate controls and corrected using DMSO only treated conditions incorporated throughout each screen. In-house software[78] was used to fit the inter-plate or intra-plate titration data with the standard Hill equation (Supplementary Equation (5)) and CRCs were classified by activity as previously described[27,79]. Curve classes of ±1.1, ±1.2, ±2.1 and ±2.2 were considered active and data was visually confirmed in Prism as described as above.

3-axes plots were generated for the firefly luciferase (FLuc) functional and SDR assay screens with the 3D qHTS Waterfall Plot software[79]. TIBCO Spotfire with PerkinElmer Lead Discovery was used to perform UMPGA hierarchical clustering of compounds by structure to identify compound clades having high similarity (Tanimoto Index > 0.8). Data from qHTS has been deposited into PubChem as detailed Supplementary Table 2.

**Protein structure models.** Visualization of protein structures employed the use of Mol* Viewer and PyMOL software[80].

All replicate data analyzed in this manuscript were for distinct samples.

**Reporting summary**

Further information on research design is available in the Nature Portfolio Reporting Summary linked to this article.

## Data availability

Previously published crystal and NMR structures used in this study 1OPK, 1YDT, 5KGL, 5KGN, 5TT5, 2OWO, 1AOI, 4M6K, 2L28 and 1AO8 were downloaded from the Protein Data Bank (www.wwpdb.org)[81]. Chemical library qHTS data sets are available from the PubChem AIDs 1963320, 1963319, 1963318, 1963317, 1963315, 1963321, 1963322 of the PubChem data base (https://pubchem.ncbi.nlm.nih.gov/)[82]. Source data are provided with this paper.

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

## Acknowledgements

This research was supported by the Intramural Research Programs of the National Center for Advancing Translational Sciences, NIH under project 1ZIATR000053 (J.I.). The content of this publication does not necessarily reflect the views or policies of the Department of Health and Human Services, nor does mention of trade names, commercial products, or organizations imply endorsement by the U.S. Government. The authors acknowledge B Callahan (Binghamton U.), J Hacia (USC), M Hall, and A Simeonov (NCATS) for critical reading of the manuscript, NCATS colleague H-L Lin and Z Itkin for customized compound-containing assays plate preparation, the Compound Management and Analytical Chemistry Departments, D Tao and Y Fang for HRMS analysis, GB Rai for fluorescein labeling of Ce-2d-FL, E Oliphant, EK Carlson, S Knibbs and T Kinder for assistance with DNA constructs and technical support. Protein structure visualization employed the use of Mol* Viewer and PyMOL software.

## Author contributions

Conceptualization: J.I. and P.K.D. Investigation: J.I., D.A.C. and P.K.D.; Data curation: P.K.D., D.A.C., J.C.B., Y.W., R.T. and M.A.; Formal analysis: J.I., P.K.D., J.C.B., R.T., D.A.C. and K.M.W.; Methodology: J.I., P.K.D., D.A.C., M.A., D.S. and J.B.; Resources: D.A.C., R.v.N., M.A., J.C.B., and K.M.W.; Validation: P.K.D. and D.A.C.; Visualization: J.I., J.C.B., and P.K.D.; Writing–original draft: J.I., D.A.C. and P.K.D.; Writing–review & editing, all authors: Supervision: J.I.; Funding Acquisition: J.I. All authors have approved the final version of the manuscript.

## Funding

## Competing interests

The authors declare no conflicting interests pertaining to the material in this manuscript.
