## [Transparent Peer Review file · Nature Communications]

A general assay platform to study protein pharmacology using ligand-dependent structural dynamics

Corresponding Author: Dr James Inglese

Version 0:

Reviewer comments:

Reviewer #2

(Remarks to the Author)

Allosteric inhibition is a novel approach to drugging specific target proteins without directly inhibiting catalytic or other activities. To identify allosteric inhibitors, however, it is always necessary to establish unique assays for every enzyme and tuning substrate, cofactor and inhibitor library concentrations. Ciulla et al. describe a general strategy to detect allosteric modulation of proteins they call alpha-complementation LIGand-103 dependent High Throughput (alpha-LIGHT). It is based on the Nanoluc luciferase protein-fragment complementation assay, in which a small 11 amino acid alpha helical fragment (HiBit) is fused to the protein of interest at N- or C-terminal domains. A complementary large 18 kD fragment is then added and luciferase activity monitored. One then detects allosteric transitions following addition of cofactors, known allosteric modulators or libraries of potential modulators as changes in luciferase activity, generally in the cases studied, increases. As a proof-of-concept target, they used the ATP-dependent firefly luciferase (FLuc). Simultaneously they directly measured FLuc activity to compare to alpha-LIGHT. They then screened a library of 1,343 FLuc inhibitory compounds showing strong correlations between allosteric effects detected by the two assays. They then investigated ability to detect allosteric activities of small molecules for kinases, isomerases, reductases, and ligases, including ABL kinase, Protein Kinase A (PKA), phosphoglycerate mutase (iPGM) two DNA ligases, and Dihydrofolate reductase. In each case they were able to demonstrate alpha-LIGHT luciferase activity modulation.

This is an exceptionally brilliant idea and very well performed study. Although it cannot be guaranteed that alpha-LIGHT will work as a detector of allostery for every enzyme the result with these diverse enzyme classes suggest that it would likely work for many enzymes and perhaps non-catalyst proteins. The authors readily admit that they do not understand why the assay works but suggest that it could be due to conformation changes or shifts in vibrational modes. I'd also speculate that it could be due to shifts in low frequency normal modes. In any case, it works. A similar strategy has been used by Ben Lehner's group, but with DHFR fragment complementation, there, the measurements are in vivo, indirect and generally used to detect abundance and stability of protein mutant variants (see e.g. Faure, A.J., et al. (2022, 2024). Nature. However, the DHFR assay couldn't be used to unambiguously measure small molecule-driven allostery in living cells, because any effects of the small molecules on DHFR activity could be direct or indirect. Lehner's group also use the DHFR assay to detect allosteric sites modulating protein-protein interactions (Weng, C., et al. (2024) Nature). The same strategy could be used with alpha-LIGHT for small molecules, a method that could be useful for non-enzyme targets.

Overall, I consider this work a highly significant advance in the study of protein allostery worthy of publication in Nature Comm. and I recommend publication with no changes.

Reviewer #3

(Remarks to the Author)

General comments:

In this paper, the authors present a novel approach using split-Nanoluciferase to monitor ligand-protein binding. Target proteins are fused with the HiBit tag, renamed alpha peptide, which interacts with the omega tag (previously known as the

Lgbit domain of Nanoluciferase) for detection. Protein-ligand binding induces conformational stabilization, leading to enhanced luminescence. After a first proof of concept with an inhibitor of Firefly luciferase, the methodology is further validated across various case studies, including protein kinases, DNA ligases and the DHFR enzyme. For this set of proteins, the authors provide a detailed comparison with existing activity assays, correlating the alpha-complementation signal with ligand binding to the target proteins. The methodology is well described, and the study offers an innovative and cost-effective approach for target proteins that lack available activity assays. However, the mechanism is intriguing and not fully addressed in the study. Controls assessing ligand binding in correlation with the alpha signal should be implemented to gain deeper insights into the mechanism and guide future applications of the system.

Major:

1- Fig. 1c: The stabilization of Fluc with PCT124 coincide with increase of alpha-complementation. If the increase of alpha correlation is linked to Fluc stabilization, then this ligand-induced stabilization should be detected using biophysical methods. Correlating the alpha signal with stability assays such as thermal shift assay (ie DSF) will support the starting hypothesis.

2- One hypothesis is that the ATP-based ligand improves complementation with alpha and omega and stabilizes the reconstituted luciferase. One simple experiment is to verify whether the alpha curves are similar when the full-length nanoluc or split-nanoluc are used (with PCT124).

3- Fig 1e. Chemodependence on ATP. Could it be possible that there is an allosteric ATP binding site in Nluc? The author should highlight the mechanism by evaluating the binding of a fluorescent ATP analog to probe the allosteric conformational change in the active site of Nluc or split-Nluc (ie: fluorescence spectroscopy or equivalent). Compounds on clade O and P likely occupy the Fluc ATP site independently of ATP, and impact further alpha complementation. Controls on the binding of these probes to either FLuc or Nluc will provide a better understanding of the mechanisms involved.

4- Fig.2 in the kinase inhibitor section Fig 2 (imatinib and H89), the lower dynamic range for alpha light is <2 those from the kinase assay? (fig 2e) How do the authors explain this behavior? Is it because NLuc luminescence is ATP-independent and there is a strong luminescence signal before addition of the ligand?

5-Fig 2h. In the PKA/regulatory subunit assay, the alpha complementation decrease could be linked to dissociation with cAMP ligand (competition). This model could be verified using an unrelated compound and control with a PPI assay version of Nluc.

6- Fig 4. illustrates nicely the dependence of the alpha assay to monitor ligand-specific binding. However, the assay varies widely on the tagging configuration of the ligases, which the authors attribute to the proximity of N and C-termini. How the alpha complementation is dependent on this topological feature since they are self-assembling? This may be related to protein folding issues or inaccessibility of alpha tag rather than proximity of C-termini and protein size. Discuss.

7- Inhibitors have been used in the study. What about agonists? Could the response of the alpha assay is expected to be different/opposite between from inhibitor/agonist? Ideally, a test case comparing an agonist to an inhibitor would expand further applications of the technique, as screening for agonists is typically more challenging than for inhibitors using classical HTS assays.

Minor:

1- This study reports the quantification of ligand binding to target proteins tagged with the Nanoluc complementation system. Since alpha corresponds to Hbit and omega corresponding to LgBit, it appears that they spontaneously associate. However, the text refers to Dixon et al. which discusses the engineering of reduced split Nanoluc assembly for protein interaction assays. It should be specified that they refer to the self-assembling domains of Nanoluc to avoid any confusion.

2- Fig2. Line216. Regarding the correlation between inhibition curves and the increase in alpha light, the term "higher sensitivity" of the alpha light signal gain compared to the PK assay is relative (factor 7/6) Given the high initial background luminescence using alpha assay, what precautions should be taken during use? (protein concentration ...)

3- Since the assay is effective only with ATP cofactor ligand binding, this should be mentioned in the abstract.

Version 1:

Reviewer comments:

Reviewer #3

(Remarks to the Author)

This revision is greatly appreciated, as the authors have made significant efforts to address my concerns accurately. They have successfully responded to all the critical points raised in the first round of review.

-An important experiment involved the correlation between the alpha light, now referred to as the SDR signal, with ligand binding properties. The authors performed Thermofluor analysis to demonstrate on selected compounds the binding to Fluc, presenting their new results in Figure S6.

-The second key point was the ATP dependence, which raised questions about the possibility of an allosteric ATP binding site in NLuc, potentially leading to a misinterpretation. In new experiments presented in Figure S5, the authors demonstrate the absence of a putative ATP binding site in NLuc.

-Finally, the authors evaluated the ligand-mediated effects on NLuc and split-NLuc bioluminescence, incorporating this information into Figure 1d, g. They also provided additional details on ligand-binding specificity, particularly ATP binding, by demonstrating a novel SDR assay using the myristate-binding kinase ABL1. Their results reveal the allosteric binding of the myristate ligand, which could not be detected using conventional kinase assays.

I therefore believe their work is ready for publication without further changes.

Reviewer #4

(Remarks to the Author)

I believe the authors addressed the comments and questions appropriately, but I have some additional suggestions and comments for improvement:

1. The authors could still improve on the readability of the manuscript by better anticipating possible confusion of the readers. At least two sources of possible confusion are that (1) the first target is itself a luciferase (so mention this explicitly) and (2) the use of a split luciferase system (which normally works by template mediated complementation in e.g. PPI), whereas the effect described here does not rely on a split enzyme system
2. All assays described report the luminescence intensity, which is well-known to be environmentally sensitive and also time-dependent (e.g. of substrate turn-over). This aspect is worth mentioning (how did the authors deal with it) as well as possible solutions (e.g. the use of a non-responsive second calibrator luciferase).
3. The reported effect on NLuc activity could be either on K_m or on k_{cat} (or both). Did the authors check whether the effect is dependent on the substrate concentration?

Reviewer #5

(Remarks to the Author)

This excellent manuscript by Ciulla et al. describes the development of a novel assay to detect ligand binding to a target protein by means of SDR, Structural Dynamics Response. The method is demonstrated by monitoring the change in NanoLuc bioluminescence (intact and in the complementing form) fused to seven different target enzymes, when bound by distinct ligands at varying concentrations. The luminescence modulations observed are shown to correlate with and be more sensitive than some other functional readouts, and the assay can be used with purified proteins or raw cell lysates. The manuscript is clear, well-written, and accessible. The authors provide exhaustive and convincing data supporting the reproducibility and utility of their new technique.

The mechanism of the SDR assay is not explained in the manuscript, beyond conjecture about "structural rearrangements and vibrational dampening". This restriction in scope is acceptable, given the potential value of the new assay to the field, and given that it will provide generous opportunities for investigation into the biophysical effect shown. The variety of motional modes and other dynamic characteristics that can be modulated by ligand binding is vast and not well understood. The contribution of protein motion to activity is well-studied in only a few model systems. The multifactorial nature of the problem could relate to the observation that the amplitude and direction of the SDR effect varied widely among the systems they tested. This merits further research.

I have no concerns about the paper, and the authors have, in my opinion, already very amply responded to the concerns raised by Reviewers #2 and #3. I recommend that the manuscript be accepted as is (although they might also wish to include RLU in the list of abbreviations).

Reviewer's Comments:

Reviewer #2 (Remarks to the Author)

Allosteric inhibition is a novel approach to drugging specific target proteins without directly inhibiting catalytic or other activities. To identify allosteric inhibitors, however, it is always necessary to establish unique assays for every enzyme and tuning substrate, cofactor and inhibitor library concentrations. Ciulla et al. describe a general strategy to detect allosteric modulation of proteins they call alpha-complementation LIGand-103 dependent High Throughput (alpha-LIGHT). It is based on the Nanoluc luciferase protein-fragment complementation assay, in which a small 11 amino acid alpha helical fragment (HiBit) is fused to the protein of interest at N- or C-terminal domains. A complementary large 18 kD fragment is then added and luciferase activity monitored. One then detects allosteric transitions following addition of cofactors, known allosteric modulators or libraries of potential modulators as changes in luciferase activity, generally in the cases studied, increases. As a proof-of-concept target, they used the ATP-dependent firefly luciferase (FLuc). Simultaneously they directly measured FLuc activity to compare to alpha-LIGHT. They then screened a library of 1,343 FLuc inhibitory compounds showing strong correlations between allosteric effects detected by the two assays. They then investigated ability to detect allosteric activities of small molecules for kinases, isomerases, reductases, and ligases, including ABL kinase, Protein Kinase A (PKA), phosphoglycerate mutase (iPGM) two DNA ligases, and Dihydrofolate reductase. In each case they were able to demonstrate alpha-LIGHT luciferase activity modulation.

This is an exceptionally brilliant idea and very well performed study. Although it cannot be guaranteed that alpha-LIGHT will work as a detector of allostery for every enzyme the result with these diverse enzyme classes suggest that it would likely work for many enzymes and perhaps non-catalyst proteins. The authors readily admit that they do not understand why the assay works but suggest that it could be due to conformation changes or shifts in vibrational modes. I'd also speculate that it could be due to shifts in low frequency normal modes. In any case, it works. A similar strategy has been used by Ben Lehner's group, but with DHFR fragment complementation, there, the measurements are in vivo, indirect and generally used to detect abundance and stability of protein mutant variants (see e.g. Faure, A.J., et al. (2022, 2024). Nature. However, the DHFR assay couldn't be used to unambiguously measure small molecule-driven allostery in living cells, because any effects of the small molecules on DHFR activity could be direct or indirect. Lehner's group also use the DHFR assay to detect allosteric sites modulating protein-protein interactions (Weng, C., et al. (2024) Nature). The same strategy could be used with alpha-LIGHT for small molecules, a method that could be useful for non-enzyme targets.

Overall, I consider this work a highly significant advance in the study of protein allostery worthy of publication in Nature Comm. and I recommend publication with no changes.

We thank the reviewer 2 for their feedback, appreciation of our work, and the discussion of allostery. We have now provided an example of the detection of an allosteric site binding molecule at the C-terminus of the ABL1 kinase domain (new Fig 2d and e) which escapes detection using the functional enzyme assay (new Fig 2f and Fig S10). This new example now provides proof-of-concept for the detection of small molecule allosteric site ligands.

Reviewer #3 (Remarks to the Author)

General comments:

In this paper, the authors present a novel approach using split-Nanoluciferase to monitor ligand-protein binding. Target proteins are fused with the Hibit tag, renamed alpha peptide, which interacts with the omega tag (previously known as the Lgbit domain of Nanoluciferase) for detection. Protein-ligand binding induces conformational stabilization, leading to enhanced luminescence. After a first proof of concept with an inhibitor of Firefly luciferase, the methodology is further validated across various case studies, including protein kinases, DNA ligases and the DHFR enzyme. For this set of proteins, the authors provide a detailed comparison with existing activity assays, correlating the alpha-complementation signal with ligand binding to the target proteins. The methodology is well described, and the study offers an innovative and cost-effective approach for target proteins that lack available activity assays. However, the mechanism is intriguing and not fully addressed in the study. Controls assessing ligand binding in correlation with the alpha signal should be implemented to gain deeper insights into the mechanism and guide future applications of the system.

We thank the reviewer 3 for their appreciation of the work and constructive criticism. We understand the interest in better defining the mechanism, and in this revision we have developed a more refined model using the reviewer's suggested experiments. Considering the results of the additional studies, we are adopting a more general term for the assay method, now referred to as the structural dynamics response (SDR) assay. We have attempted to carefully address the questions below which have permitted a fuller mechanistic understanding of the approach. Also, we now demonstrate that SDR can detect small molecule allosteric ligands.

Major:

1- Fig. 1c: The stabilization of Fluc with PCT124 coincide with increase of alpha-complementation. If the increase of alpha correlation is linked to Fluc stabilization, then this ligand-induced stabilization should be detected using biophysical methods. Correlating the alpha signal with stability assays such as thermal shift assay (ie DSF) will support the starting hypothesis.

AU: We chose FLuc as the exemplar protein to illustrate the α LIGHT (SDR) concept because we have extensively studied the chemical biology associated with this broadly used 'reporter gene'. We have previously published the thermal stability of FLuc with PTC124 \pm ATP in Ref 25 (Auld et al. 2010, Fig. 4 and Fig. S6B). Indeed, FLuc stability is increased as the concentration of PTC124 approaches a stoichiometric proportion with FLuc, and the stabilization in the presence of saturating ATP is significantly more stabilizing, mirroring our SDR findings in **Figure 1d** of the revised manuscript.

As requested by reviewer 2 we have performed thermal shift assays choosing the ThermoFluor technique to correlate the SDR response. We used PTC124 \pm ATP as a control ligand and several newly identified ligands from clade S. As show in the new data set (**Fig. S6**), we observe a strong correlation of the this ATP-dependent chemotype with thermofluor for the SAR series, which was selected to cover both active and inactive clade S analogs (see response to Major question #3 for more about clade S).

2- One hypothesis is that the ATP-based ligand improves complementation with alpha and omega and stabilizes the reconstituted luciferase. One simple experiment is to verify whether the alpha curves are similar when the full-length nanoluc or split-nanoluc are used (with PCT124).

AU: The reviewer makes an excellent point. To address this, we prepared a plasmid with the FLuc sequence followed by full length NLuc sequence, expressed and purified this new protein (FLuc-C-

NLuc). We observed that the intact NLuc can generate an α LIGHT (SDR) response to PTC124+ATP (new **Fig 1c**). This finding indicates that the ligand-mediated effect on NLuc bioluminescence is not necessarily an effect on α -complementation, rather an effect on the catalytic activity of NLuc, which appears to be more impacted in a split NLuc than the intact NLuc. In other words, a ligand's response max. is higher using the HiBiT version (see also new **Fig. 1f and g**)

We observed that the basal NLuc signal of FLuc-C-NLuc was substantially higher than that of FLuc-C-HiBiT•LgBiT, which is not surprising given the compromised structure of the split NLuc. When tested under similar basal (no ligands) assay conditions, the FLuc-C-HiBiT had ~500 reduction in RLU compared to the FLuc-C-NLuc protein. To obtain a useful S:B we need to use the FLuc-C-NLuc at 1 nM, however this gave an overall lower response efficacy than the FLuc-C-HiBiT used at 10 nM. While FLuc-C-NLuc can detect ligand binding the ability to detect a weak ligand is more reliable with the HiBiT version (see new **Fig. 1c vs d, and 1f and g**).

3- Fig 1e. Chemodependence on ATP. Could it be possible that there is an allosteric ATP binding site in NLuc? The author should highlight the mechanism by evaluating the binding of a fluorescent ATP analog to probe the allosteric conformational change in the active site of NLuc or split-NLuc (ie: fluorescence spectroscopy or equivalent).

AU: We have used several sensitive approaches including analytical size exclusion chromatography (see analytical SEC data in the appendix of this letter) to determine if ATP interacts with NLuc and find no evidence for this. We further explored the chemo-dependence of ATP on several clade S members using ThermoFluor and *P. pyralis* FLuc to establish that the ATP dependence is indeed a consequence of ATP binding to FLuc and not NLuc (see new **SI Fig S6**).

As part of our response to the above question, 'Could it be possible that there is an allosteric ATP binding site in NLuc?', we tested Clade S with NLuc enzyme expressed and purified from *E. coli*. We demonstrated high bioluminescent activity of the NLuc prep using the Nano-Glo reagent from Promega (which is a formulation of the furimazine substrate). As shown in now Supplemental **Figure S5**, none of the compounds in expanded Clade S affect NLuc activity either in the absence or presence of ATP, i.e., ATP had no effect on NLuc activity. We interpret these data to support that there is not an ATP binding site on NLuc.

We believe the most likely explanation for the ATP-dependent α LIGHT (SDR) activity of Clade S arises from a synergistic mechanism. Given the substructure (blue outline) of the Clade S core has similarity to the FLuc substrate luciferin, we can speculate that Clade S binds to the luciferin pocket when ATP is present.

Compounds on clade O and P likely occupy the FLuc ATP site independently of ATP, and impact further alpha complementation. Controls on the binding of these probes to either FLuc or NLuc will provide a better understanding of the mechanisms involved.

AU: We believe that clade O and P compounds likely occupy the FLuc ATP site, particularly clade P

which contains a purine analog heterocycle. They certainly bind to FLuc as they potently inhibit the FLuc enzymatic assay and do not have any effect on NLuc \pm ATP (solid black circles and open/solid red circles, respectively, in **Figure 1f** of the revised manuscript).

We believe that since clade O and P compounds likely occupy the same site as ATP, the α LIGHT (SDR) signal is simply responding to high affinity of the clade O or P ligand which fills the binding site and thus ATP has no additional α LIGHT (SDR) response, as it is effectively blocked from binding to FLuc.

4- Fig.2 in the kinase inhibitor section Fig 2 (imatinib and H89), the lower dynamic range for alpha light is <2 those from the kinase assay? (fig 2e) How do the authors explain this behavior? Is it because NLuc luminescence is ATP-independent and there is a strong luminescence signal before addition of the ligand?

AU: There is no question that the α LIGHT (SDR) assay has a lower dynamic range than assays that rely on the enzymatically driven accumulation of a measured product as in the kinase enzyme assay (**Fig 2**) or iPGM coupled enzyme assay (**Fig 3**). We agree with the reviewer that the lower dynamic range is in part because there is background ATP-independent luminescence from the apo-protein-HiBiT/LgBiT basal luminescence. We speculate the α LIGHT (SDR) assay concept has gone unnoticed for so long because the response, if observed was not considered significant in screening strategies using a single test concentration of library compound, however in the context of concentration response curves the SDR response is clear.

5-Fig 2h. In the PKA/regulatory subunit assay, the alpha complementation decrease could be linked to dissociation with cAMP ligand (competition). This model could be verified using an unrelated compound and control with a PPI assay version of NLuc.

AU: Upon further consideration we have decided to remove the panels describing the PKA/Regulatory subunit experiment to focus the paper on target protein ligand binding where the SDR is directly observed. We consider protein-protein interactions beyond the scope of this initial communication.

6- Fig 4. illustrates nicely the dependence of the alpha assay to monitor ligand-specific binding. However, the assay varies widely on the tagging configuration of the ligases, which the authors attribute to the proximity of N and C-termini. How the alpha complementation is dependent on this topological feature since they are self-assembling? This may be related to protein folding issues or inaccessibility of alpha tag rather than proximity of C-termini and protein size. Discuss.

AU: We agree there may be a variety of explanations for these differences in α LIGHT (SDR) output response beyond our suggestion of N- and C-termini proximity to the ligand binding site, for example, as the reviewer points out. We have included the following sentence to include these possibilities at the end of the first paragraph of the '*DNA Ligases*' section: 'This observed assay response variation, in both magnitude and direction, may be a consequence of DNA-mediated domain motions either exposing or obscuring the terminal α -peptide sequence, or affect the split NLuc confirmation.'

7- Inhibitors have been used in the study. What about agonists? Could the response of the alpha assay is expected to be different/opposite between from inhibitor/agonist? Ideally, a test case comparing an agonist to an inhibitor would expand further applications of the technique, as screening for agonists is typically more challenging than for inhibitors using classical HTS assays.

AU: The closest example we have in this study that the α LIGHT (SDR) assay might detect 'agonists' is our observation that we can detect substrate binding, for example ATP, NAD⁺, or DNA oligo, suggesting alternate substrates might be identified using a system such as α LIGHT (SDR). We now

also show how the method can detect allosteric ligands to expand the range of applications (new **Fig. 2d** and **e**). Extending the α LIGHT (SDR) method to proteins such as GPCRs might be possible, but we believe it would be beyond the scope of this initial study.

Minor:

1- This study reports the quantification of ligand binding to target proteins tagged with the Nanoluc complementation system. Since alpha corresponds to Hibit and omega corresponding to LgBit, it appears that they spontaneously associate. However, the text refers to Dixon et al. which discusses the engineering of reduced split Nanoluc assembly for protein interaction assays. It should be specified that they refer to the self-assembling domains of Nanoluc to avoid any confusion.

AU: We thank the reviewer for pointing this out. Yes, we are using the general α -complementation definition for the two components as originally described by Ullmann et al (1967). To implement α LIGHT (SDR) we are using Promega self-assembling HiBiT and LgBiT domains for the reporter component. We have edited line 113 to now read, 'For initial characterization of the SDR concept, either NLuc or the 11 amino acid HiBiT tag (α peptide) and complementary 18 kDa LgBiT subunit (ω fragment), which self-assembles to reconstitute functional NLuc enzyme (Dixon et al ref) were tested'.

2- Fig2. Line216. Regarding the correlation between inhibition curves and the increase in alpha light, the term "higher sensitivity" of the alpha light signal gain compared to the PK assay is relative (factor 7/6)

Given the high initial background luminescence using alpha assay, what precautions should be taken during use? (protein concentration ...)

AU: The factor of $pSDR_{50}$ of 7/ pIC_{50} of 6 is a 10-fold difference, and is dependent on assay specific factors, so it is relative, but nonetheless illustrates a rather sensitive assay, primarily because the α LIGHT (SDR) assay can be measured in the absence of a potentially competing cofactor.

We have expanded this paragraph to provide more context and background, 'Correlated inhibitor potencies obtained from the functional versus SDR assays illustrate, aside from a few exceptions, an equivalence or apparent higher sensitivity of the gain-of-signal SDR assay format (**Figure 2c**). Assay sensitivity or robustness will be, in part, based on factors such as the quantifiable response for a specific target protein concentration and the ligand binding mechanism, e.g., ortho- vs allosteric site binding. Also, target protein concentrations may exceed the K_D of potent ligands limiting the observable affinity (e.g., see **Table 1**). This could, for example, obscure rank ordering of potency in a medical chemistry lead optimization series, but is significantly less likely to impact discovery of nascent unoptimized compounds typically having K_D values well above the low nM target protein assay concentration³⁴'

3- Since the assay is effective only with ATP cofactor ligand binding, this should be mentioned in the abstract.

AU: We are not clear as to what is being referred to as 'effective only with ATP cofactor ligand binding'. We will respond to several possible interpretations. The α LIGHT (SDR) assay does not require ATP, and any ATP-dependent enzyme tested using the method in this study, e.g. the firefly luciferase, protein kinases or the T7 ligase did not require ATP for α LIGHT (SDR) to detect ligand binding.

OR is the suggestion that the α LIGHT (SDR) method as demonstrated with several ATP utilizing enzymes, only finds ATP binding site ligands? To address this, we took advantage of the allosteric

myristate binding site at the *C*-terminus of the ABL1 kinase domain protein (new **Fig 2a**). Using the ABL1-*N*-HiBiT and a newly constructed ABL1-*C*-HiBiT we demonstrate that the α LIGHT (SDR) assay can detect asciminib, a specific myristate pocket ligand. We show that the ABL1-*C*-HiBiT (new **Fig 2d**) has better detection sensitivity likely owing to the proximity of the NLuc sensor to the allosteric pocket compared to the ABL1-*N*-HiBiT (new **Fig 2e**) which also detects asciminib, but with lower efficacy. Importantly, we show that an assay based on kinase activity does not detect asciminib (new **Fig S10**), therefore illustrating the advantage of the SDR assay for detection of allosteric site ligands that have no impact on catalytic activity.

Letter Appendix -analytical SEC data

Size exclusion chromatography to investigate potential ATP binding site on NLuc

Experimental. Enzymes, C.e. iPGM (a), FLuc (d) or NLuc (g) and fluorescein-labeled compounds Ce-2d-fluorescein (Ce-2d-FL, b) or ATP-12-fluorescein (ATP-FL, e and h; Revvity, Cat # NEL439001EA) were prepared at 50 μ M in storage buffer (125 mM Tris-Cl, 25 mM Mg₂SO₄, and 100 mM KCl) and mixed in a 1:1 ratio in a 1.5 mL amber-colored centrifuge tube, then incubated at room temperature for 10 minutes. A 0.1 mL aliquot of the 'complex', (c) iPGM•Ce-2d-FL, (f) FLuc•ATP-FL or (i) NLuc•ATP-FL was loaded onto a Superdex 75 3.2/300 analytical size exclusion column (Cytiva, Cat # 29036230) with a bed volume of 2.4 mL, using a 10 μ L sample loop, and eluted with storage buffer at a flow rate of 0.1 mL/min. The elution profile was monitored using a UV-Vis absorption detector at wavelengths of 280 nm (blue trace) and 500 nm (purple trace), and fluorescence (EX480/EM520) was detected using a Shimadzu RF-20A/RF-20Axs fluorescence detector to assess the interaction between the enzyme and labeled compound. The fluorescence trace (red) lags the absorbance (blue/purple) due to the detector's position after the absorbance detector.

REVIEWERS' COMMENTS

Reviewer #3 (Remarks to the Author)

This revision is greatly appreciated, as the authors have made significant efforts to address my concerns accurately. They have successfully responded to all the critical points raised in the first round of review.

-An important experiment involved the correlation between the alpha light, now referred to as the SDR signal, with ligand binding properties. The authors performed Thermofluor analysis to demonstrate on selected compounds the binding to Fluc, presenting their new results in Figure S6.

-The second key point was the ATP dependence, which raised questions about the possibility of an allosteric ATP binding site in NLuc, potentially leading to a misinterpretation. In new experiments presented in Figure S5, the authors demonstrate the absence of a putative ATP binding site in NLuc.

-Finally, the authors evaluated the ligand-mediated effects on NLuc and split-NLuc bioluminescence, incorporating this information into Figure 1d, g. They also provided additional details on ligand-binding specificity, particularly ATP binding, by demonstrating a novel SDR assay using the myristate-binding kinase ABL1. Their results reveal the allosteric binding of the myristate ligand, which could not be detected using conventional kinase assays.

I therefore believe their work is ready for publication without further changes.

AU: We thank R3 again for their constructive suggestions which has significantly refined our thinking on the SDR mechanism.

Reviewer #4 (Remarks to the Author)

I believe the authors addressed the comments and questions appropriately, but I have some additional suggestions and comments for improvement:

1. The authors could still improve on the readability of the manuscript by better anticipating possible confusion of the readers. At least two sources of possible confusion are that (1) the first target is itself a luciferase (so mention this explicitly) and (2) the use of a split luciferase system (which normally works by template mediated complementation in e.g. PPI), whereas the effect described here does not rely on a split enzyme system
2. All assays described report the luminescence intensity, which is well-known to be environmentally sensitive and also time-dependent (e.g. of substrate turn-over). This aspect is with mentioning (how did the authors deal with it) as well as possible solutions

(e.g the use of a non-responsive second calibrator luciferase).

3. The reported effect on NLuc activity could be either on K_m or on k_{cat} (or both). Did the authors check whether the effect is dependent on the substrate concentration?

AU: We thank R4 for their suggestions.

1- point 1, in the first line of the Results we now state, ‘... to either the N- or C-terminus of the target protein, firefly luciferase, an ATP-dependent...’, to more clearly indicated that FLuc is the target protein (pg. 4, line 109). Also, on pg. 4 line 114, we now indicated, “In contrast to the ATP-dependent bioluminescent FLuc enzyme, NLuc generates a bright bioluminescence in an ATP-*independent* manner from the direct oxidation of furimazine.” Finally, we have added the sentence, “Therefore, these SDR constructs allow us to measure ligand binding to FLuc without a reliance on FLuc enzyme activity” to further emphasize that we are not using FLuc luminescence to measure the ligand binding to FLuc.

1- point 2, we have indicated in several places within the manuscript that the nature of the α -complementation used here is ‘self-assembling’ (pg. 4, line 99), ‘high-affinity’ (pg. 4, line 99), ‘which self-assemble to reconstitute functional NLuc enzyme²⁴’ (pg. 4, line 113). To further clarify we are using the α -complementation tag not intended for PPI detection, we replaced ref 5 (Dixon) and cite Schwinn et al. 2018 (new ref 24) which describes the HiBiT tag and its use for directly quantifying proteins (pg. 4 line ~10 and line ~113). We reserved the use of Dixon et al (ref 5) for our discussion in paragraph 1 of the introduction.

2-Because we are not attempting to compare the RLU of one target protein to another we do not require a non-responsive second calibrator luciferase. We rely on the DMSO control for the basal level of target-NLuc or target-HiBiT/LgBiT signal and compare this to the ligand bound complex RLU. Figure S10 illustrates that the basal RLU differs between target-HiBiT/LgBiT fusion proteins. As the reviewer points out this is likely due to optimized buffer conditions specific to each target-fusion protein.

3- To address the question of whether the SDR could arise from an effect K_m or k_{cat} of NLuc we conducted an experiment with FLuc-C-NLuc varying the NLuc substrate concentration in the presence or absence of a near saturating concentration of an FLuc ligand. Accordingly, this preliminary investigation into the effect of SDR on NLuc enzyme kinetics suggests both an enhancement of V_{max} observed during the pre-steady state phase (to extent we could measure this phase), and an attenuated loss of signal during the steady-state phase (see the **Appendix** at the end of this response). This latter phenomenon may be attributed to SDR influencing the degree of product inhibition previously reported by Nemergut et al. (Nature Communications, 2023). While this warrants further investigation, we believe this would be beyond the scope of our initial assay development study.

Reviewer #5 (Remarks to the Author)

This excellent manuscript by Ciulla et al. describes the development of a novel assay to detect ligand binding to a target protein by means of SDR, Structural Dynamics Response. The method is demonstrated by monitoring the change in NanoLuc

bioluminescence (intact and in the complementing form) fused to seven different target enzymes, when bound by distinct ligands at varying concentrations. The luminescence modulations observed are shown to correlate with and be more sensitive than some other functional readouts, and the assay can be used with purified proteins or raw cell lysates. The manuscript is clear, well-written, and accessible. The authors provide exhaustive and convincing data supporting the reproducibility and utility of their new technique.

The mechanism of the SDR assay is not explained in the manuscript, beyond conjecture about "structural rearrangements and vibrational dampening". This restriction in scope is acceptable, given the potential value of the new assay to the field, and given that it will provide generous opportunities for investigation into the biophysical effect shown. The variety of motional modes and other dynamic characteristics that can be modulated by ligand binding is vast and not well understood. The contribution of protein motion to activity is well-studied in only a few model systems. The multifactorial nature of the problem could relate to the observation that the amplitude and direction of the SDR effect varied widely among the systems they tested. This merits further research.

I have no concerns about the paper, and the authors have, in my opinion, already very amply responded to the concerns raised by Reviewers #2 and #3. I recommend that the manuscript be accepted as is (although they might also wish to include RLU in the list of abbreviations).

AU: We thank R5 for their appreciation of our efforts to demonstrate the application of the approach and their perspective on the mechanistic possibilities of the phenomenon. We have added RLU to the abbreviation list.

Appendix

FLuc-NLuc Initial Velocity of SDR Effect

	DMSO	PTC124 + ATP
V_{max} (RLU s ⁻¹)	368,000 ± 12,000	668,000 ± 28,000
K_m (%Nano-glo reagent)	12.1 ± 0.9	20.5 ± 2.1

Preliminary kinetic analysis of SDR effect on Nanoluciferase catalysis. FLuc-C-NLuc was prepared in FLuc assay buffer (see **Methods**) at 25 pM final concentration and incubated for 30 min with either DMSO or 10 μ M of PTC124 + 10 μ M ATP. Enzyme solution was then added to a 96-well white flat bottom Costar plate and luminescence was read using TECAN Spark plate reader (5 s read interval, 100 ms integration time). A 2.5-min preincubation phase was measured prior to Nano-glo Reagent addition to initiate NLuc catalysis. Nano-glo reagent was serially diluted from 10% final concentration to 0.04%, 1:2, 9-pt. Error bars represent SD, n = 3. Michaelis-Menten plot was determined from the kinetic profiles. The initial rate was determined by fitting the pre-

steady state phase to linear regression and plotted against the relative Nano-glo Reagent concentration. V_{max} and K_M were determined by fitting replicate data to Initial Velocity = $V_{max} * [\text{Nano-glo Reagent}] / (K_M + [\text{Nano-glo Reagent}])$.